# Brain Treebank: Large-scale intracranial recordings from naturalistic language stimuli

**Christopher Wang**[*1,2], **Adam Yaari**[*1,2], **Aaditya K Singh**[1,5],
**Vighnesh Subramaniam**[1,2], **Dana Rosenfarb**[1,2], **Jan DeWitt**[1], **Pranav Misra**[2,3,4],
**Joseph R Madsen**[4], **Scellig Stone**[4],
**Gabriel Kreiman**[2,3,4], **Boris Katz**[1,2], **Ignacio Cases**[1,2], and **Andrei Barbu**[1,2]

[1]Computer Science and Artificial Intelligence Laboratory, MIT
[2]Center for Brains, Minds and Machines, MIT
[3]Center for Brain Science, Harvard University
[4]Boston Children's Hospital, Harvard Medical School
[5]Gatsby Computational Neuroscience Unit, University College London

## Abstract

We present the Brain Treebank, a large-scale dataset of electrophysiological neural responses, recorded from intracranial probes while 10 subjects watched one or more Hollywood movies. Subjects watched on average 2.6 Hollywood movies, for an average viewing time of 4.3 hours, and a total of 43 hours. The audio track for each movie was transcribed with manual corrections. Word onsets were manually annotated on spectrograms of the audio track for each movie. Each transcript was automatically parsed and manually corrected into the universal dependencies (UD) formalism, assigning a part of speech to every word and a dependency parse to every sentence. In total, subjects heard over 38,000 sentences (223,000 words), while they had on average 168 electrodes implanted. This is the largest dataset of intracranial recordings featuring grounded naturalistic language, one of the largest English UD treebanks in general, and one of only a few UD treebanks aligned to multimodal features. We hope that this dataset serves as a bridge between linguistic concepts, perception, and their neural representations. To that end, we present an analysis of which electrodes are sensitive to language features while also mapping out a rough time course of language processing across these electrodes. The Brain Treebank is available at `https://BrainTreebank.dev/`.

## 1   Introduction

A single theory of language understanding that encompasses how our brains process language, how linguists understand language, and how machines process language is still beyond our reach. Despite numerous attempts to understand how the brain processes language through investigations of compositionality [1–4], semantic categories [5–7], and surprisal [8–12], a mechanistic understanding is still lacking. Our hypothesis is that this is in part because studies often focus on small data regimes, since gathering large-scale neural recordings is extremely laborious. Yet NLP and ML research in general has shown that scale matters. In particular, even probing experiments on artificial networks require a fairly large scale to produce reliable results, certainly larger than a few hundred sentences [13, 14]. NLP would not have progressed without large-scale resources, so to enable the same kind of progress, we collect a new large-scale neuroscience dataset, which has naturalistic stimuli, is

---

* Equal contribution. Correspondence to CW at `czw@mit.edu`.

38th Conference on Neural Information Processing Systems (NeurIPS 2024) Track on Datasets and Benchmarks.

| Data | Quantity | Data | Quantity |
|---|---|---|---|
| Total subjects | 10 | Total sentences | 38,572 |
| Total hours | 43.5 | Unique sentences | 30,244 |
| Total electrodes | 1,688 | Avg. words per sentence | 6.5 |
| Avg. electrodes per subject | 16 | Total words | 223,068 |
| Total movies | 21 | Unique words | 12,412 |
| Unique movies | 26 | Unique speakers | 937 |
| Number of scenes | 46,935 | Unique part of speech labels | 17 |

Table 1: Quantitative overview of Brain Treebank

multimodal, and uses intracranial recordings — a high-spatial and high-temporal resolution recording method.

The Brain Treebank is foremost a treebank, like the Penn Treebank, and is annotated in the universal dependencies (UD) format. What distinguishes it, is that it is accompanied by both multimodal annotations and by neural recordings collected from 10 subjects who heard 223,068 annotated words while they watched Hollywood films. Subjects watched a total of 26 films (43.5 hours) as data was recorded from a total of 1,688 electrodes. To this, we add manual and automated annotations.

**Scene labels**: every scene in the movie was labeled according to the Places365 schema, [15], resulting in 46,935 scenes total. **Word onsets and offset**: while automatic speech recognition performs acceptably, errors are common, which were manually corrected. In addition, automated systems are simply not trained to offer extreme accuracy, at the millisecond level, when determining the start and end of words. Word onsets had to be manually annotated on spectrograms for every word to ensure alignment with the neural recordings. **Part of speech (POS) tags and parses**: Sentences were automatically parsed into the Universal Dependencies framework and then each part of speech tag and dependency relationship was manually corrected. While POS tagging is fairly accurate, numerous parser errors existed. **Speaker identity**: A unique identifier, which can be traced back to a given character, was given to every speaker in every movie. This was done manually, as no automated system exists to do so with any reasonable accuracy. Finally, we also curated a list of 16 automated video, audio, and language features that we provide to save computing time (see Table 4). We release all our data with a Creative Commons Attribution 4.0 International (CC BY 4.0) license.

Large scale stimuli for the neuroscience of language and multimodal understanding can enable natural experiments: the kind of post-hoc analysis of large-scale datasets that has propelled NLP and machine learning in general forward. In the long term we hope that treebanks such as ours, coupled with neural recordings, will help the creation of theories of language understanding that span linguistics, neuroscience, and NLP. To demonstrate the utility of the dataset, in addition to providing the raw data, we also take new steps toward understanding language in the brain; our contributions are:

1. A dataset of intracranial recordings across 26 different movie viewings (43.5 hours total).
2. Localization of electrode positions and alignment with common brain atlases.
3. Multiple layers of manual annotations to enable numerous experiments: scene labels, word onsets and offsets, part of speech tags, parses in universal dependencies format, and speaker identity.
4. Multiple automated annotations for 16 other language, audio, and visual features.
5. Quantitative results that show neural responsiveness to word onset and differential activation based on the position of a word within a sentence.

## 2   Related work

Previous works have studied language processing in the context of Magnetoencephalography (MEG) [17, 18], Electroencephalography (EEG) [19, 20], and functional magnetic resonance imaging (fMRI) [21–26]. In this work, we present Stereoelectroencephalography (sEEG) data with both high temporal resolution and naturalistic stimuli.

Recording the brain's response to naturalistic stimuli is critical to neuroscientific progress [27]. There exist fMRI datasets for naturalistic speech [28–30], vision [31, 32], and movies [33, 34]. And similar data has been collected for the EEG modality (speech [35], vision [36], and movies [37]). There are also movie datasets that cover both modalities [38]. However, when it comes to intracranial recordings, which provide better temporal resolution, but require invasive surgery to implant probes,

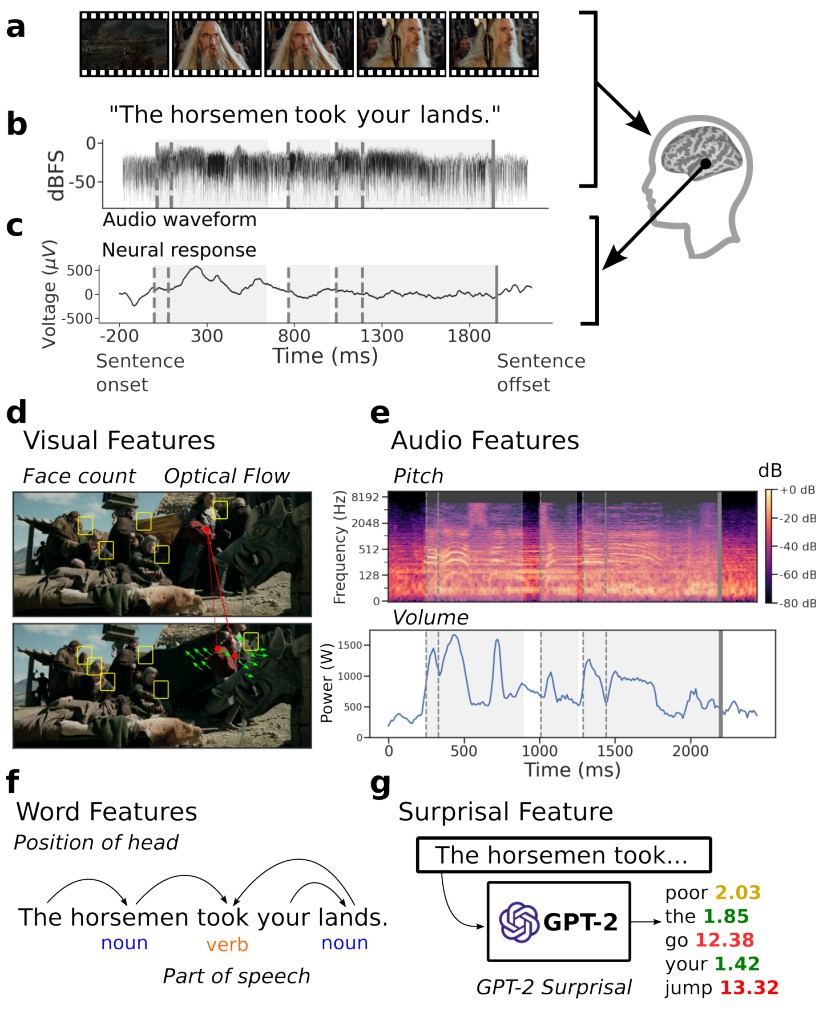

Figure 1: **Schematic of the approach. Top**: A film (**a-b**) was presented as visual and audio stimulus to the subject. Invasive neural recordings were performed while subjects watched the movie. A transcript of speech in the film is aligned to both the audio (**b**) and neural (**c**) signals. Shown here is a short signal segment from an exemplar electrode in the left superior temporal gyrus aligned to sentence onset at $t = 0$ ms. Word locations are shown as shaded regions between dashed lines. **Bottom**: Schematic overview of selected visual (**d**), audio (**e**), and language (**f-g**) features used for the General Linear Model (GLM) for each word. See Table 4 for a full list and description of features. Visual features (**d**) include the number of faces (yellow boxes), and the magnitude and angle of optical flow (green arrows). Audio features (**e**) include the average pitch (top) and volume (bottom) during each word (shaded gray regions between dashed lines). Word features (**f**) include part-of-speech and the position of each word's dependency head. A surprisal feature, (**g**), computed using GPT-2 [16], a large language model, is the negative log probability of the word given the preceding context.

data is much more sparse. There exist intracranial datasets for pose [39], speech production [40], and parts of speech [41], but none of these involve the complex natural language and concomitant visual inputs available from movie stimuli. The most similar work to our dataset, Berezutskaya et al. [42], presents participants with vastly less stimuli: a 6.5 minute short movie, compared to our average of 4.3 hours of movie per patient.

Already, the brain-recordings themselves, without annotation have proven useful for representation learning, as we show in Wang et al. [43]. And combined with the transcribed audio tracks, these have allowed for successful study of multimodal integration in the brain, as we show in Subramaniam et al. [44]. Now, for the first time, we release the complete annotated recordings for all subjects, as well as the accompanying Universal Dependency parse trees.

## 3  Data

**Dataset construction** Stereoelectroencephalography (sEEG) neural recordings were collected from 10 subjects (5 male, 5 female), aged 4-19 years ($\mu \approx 11.9$, $\sigma \approx 4.6$), under treatment for epilepsy at Boston Children's Hospital (BCH); see supplementary Table 2 for per-subject statistics. All subjects were implanted with intracranial electrodes to localize seizure foci for potential surgical resection. All experiments were approved by BCH/Harvard IRB and were carried out with the subjects' informed consent. IRB documents are available upon request, but are otherwise sensitive. Electrode types, number, and position were driven solely by clinical considerations. Recorded data was anonymized, and identifying patient information was redacted.

**Task and stimuli** Stimuli consisted of 21 recent animated/action Hollywood movies; see supplementary Table 3 for per-movie statistics. On average, movies were 2.07 hours long ($\sigma \approx 0.68$) and contained 1,443 sentences ($\sigma \approx 333$), 8,966 total words ($\sigma \approx 2068$), 1,749 unique words ($\sigma \approx 315$), 1,328 unique lemmas ($\sigma \approx 251$), 1,218 nouns ($\sigma \approx 271$), 610 unique nouns ($\sigma \approx 126$), 1,343 verbs ($\sigma \approx 293$), and 508 unique verbs ($\sigma \approx 98$). Each subject was given a choice of which movies to watch, viewing an average of 2.6 movies ($\sigma \approx 1.7$) corresponding to 4.3 hours ($\sigma \approx 3.6$). For further details, see Appendix A.3.

**Data acquisition and signal processing** Clinicians implanted subjects with intracranial stereo-electroencephalographic (sEEG) depth probes containing 6-16 0.8 mm diameter 2 mm long contact electrodes recording Intracranial Field Potentials (IFPs). Each subject had multiple (12 to 18) such probes implanted in locations determined by clinical concerns entirely unrelated to the experiment, informed by a functional analysis [45]. The number of electrodes per subject ranged between 106 and 246 ($\mu \approx 167$, $\sigma \approx 38$) for a total of 1,688 total electrodes; see supplementary Table 2 for a per-subject breakdown. Data collected during periods of seizures or immediately following a seizure was discarded. For each electrode, a notch filter was applied at 60 Hz and harmonics before analysis. No other processing (downsampling, filtering specific frequency bands, etc.) was performed on the neural recordings. For further details, see Appendix A.4. Finally, the location of all electrodes was identified and mapped to the common brain atlases (Desikan et al. [46] and Destrieux et al. [47]). For the purposes of region analyses, electrodes in white matter are projected to the grey-white boundary and assigned to the closest atlas region. Region analyses in this paper are given with respect to the Desikan-Killiany atlas. See Appendix A.5 for further details.

**Audio transcription and alignment** For each movie, the timestamps for all words in the audio were transcribed and timestamps for each word were found programmatically and then manually corrected by trained annotators (see Appendix A.1 for further details). The pipeline developed for this audio transcription and alignment effort is an independently useful source of annotated stimuli, which can now be used for further experiments. We described this pipeline more completely in a separate technical paper: Yaari et al. [48]. Part of speech tags and dependency parses were manually corrected and speaker identity and scene labels were manually annotated from scratch by an in-house expert hired at MIT.

**Feature annotation** To model the neural responses during the complex movies, we considered a series of 16 features (Table 4). These features include 5 visual attributes (pixel brightness, global optical flow magnitude, optical flow magnitude, optical flow angle, and number of faces, Figure 1d), 4 auditory attributes (volume, pitch, delta volume, and delta pitch, Figure 1e), and 6 language attributes (GPT-2 surprisal, word time length, word time difference, index in sentence, word head, and part of speech tag, Figure 1f-g). All of these features were aligned to and computed for each word. Table 4 provides a brief description of each feature, and their calculation is described in Appendix A.2. Additionally, scenes and speakers were labeled for each movie. Scenes were extracted based on camera cuts using PySceneDetect [49]. Each scene was labeled based on the corresponding image environment and labels were extracted from the Places365 dataset [15]. Finally, for each sentence in the audio transcript, the speaker identity was manually annotated (see Appendix A.2).

## 4  Quantitative analyses of language function with the dataset

**Word onsets triggered strong neural responses** After aligning the neurophysiological data to the occurrence of words (Figure 1a-c), we assessed whether the neural responses were modulated by word onset by comparing the mean activity in 5 pairs of consecutive windows of 100ms duration

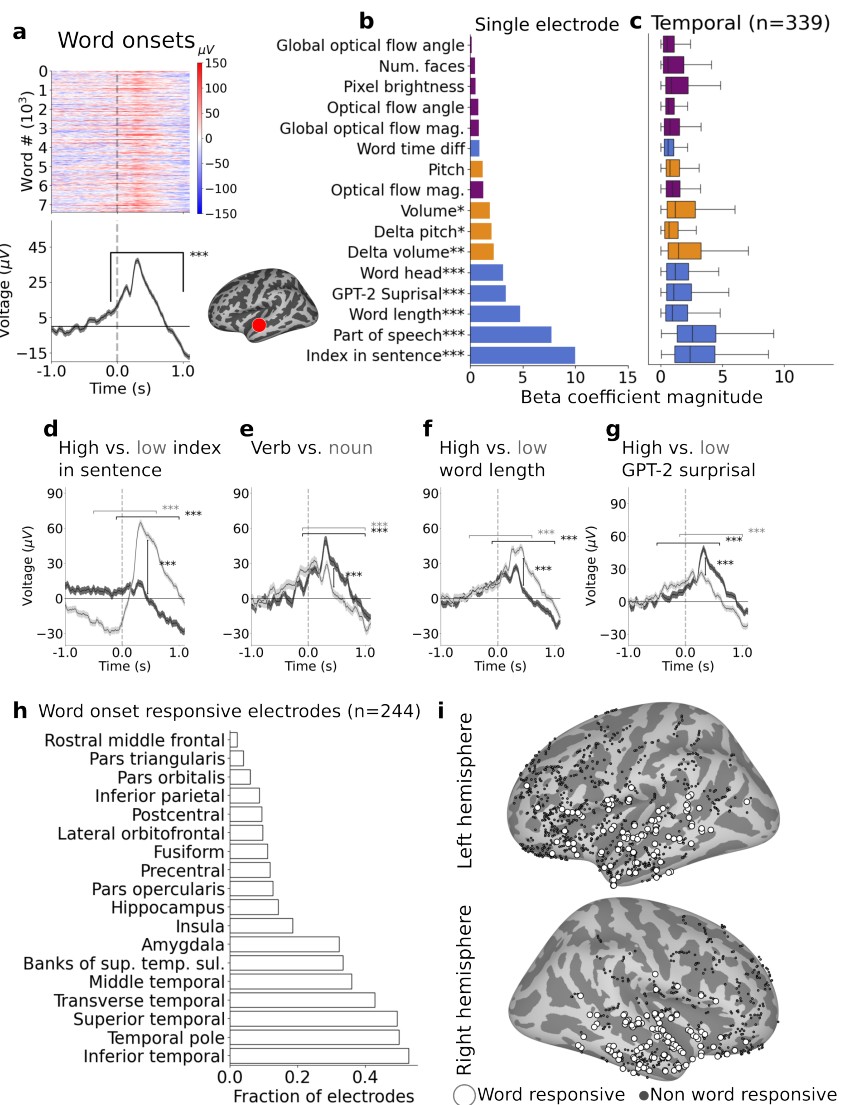

Figure 2: **Alignment to word onsets reveals strong neural responses. a.** Raster (top) and mean (bottom) plots of neural activity aligned to word onsets ($t = 0$ ms) for an exemplar electrode (inset; shown in red) in the left superior temporal sulcus. Each line in the raster is a separate word ($> 6,000$ words) in the movie. Shading in the mean plot indicates standard error. Asterisks indicate the significance (double-tailed paired t-test) of the response, measured by comparing mean activity in pre- and post- word-onset intervals (see Section 4). A GLM was fitted to predict the average response in the 500ms window after word onset (Section 4). The magnitude of the beta coefficients for all features is shown for the same example electrode (**b**) and averaged across all electrodes in the temporal lobe (**c**). Features are shown colored by category (blue: language, orange: audio, purple: visual). Asterisks indicate statistical significance of the beta coefficient for the example electrode (see Section 4). Neural responses are shown from the same example electrode separated by (**d**) index in sentence, (**e**) part of speech, (**f**) word length, (**g**) GPT-2 surprisal. Asterisks on horizontal brackets indicate the significance of the neural response, i.e., the difference between pre- and post- word-onset activity, as in (**a**). Vertical brackets show the differences in mean sub-sampled activity (see Section 4). In **h**, the fraction of electrodes per regions for which a significant word-onset response can be observed even after sub-sampling for visual and audio features is shown. The precise location of these electrodes is shown in **i**.

before (-500 ms to 0 ms) versus after (500 to 1000ms) word onset. We defined an electrode to be *word-responsive* if it yielded a statistically significant difference in at least one of the 5 pairs (paired t-test, p<0.05, Bonferroni corrected, see Appendix A.6). Figure 2a shows the neural responses of an example electrode located in the left superior temporal sulcus (Figure 2a inset). The raster plot (top) and average activity (bottom) show strong activation triggered by the onset of each word. This activity can be readily appreciated for almost every word in the more than 7,000 words (raster plot) of a single movie. Interestingly, the activity of this electrode begins to show a slight deviation from baseline *before* the onset of words at time 0. The complex nature of natural stimuli like the movie implies that multiple variables could in principle drive the neural responses. Indeed, the responses to individual words in Figure 2a show a strong degree of heterogeneity. To gain insight into what could drive these diverse responses, we considered a set of 16 visual, auditory, and language features (Table 4, Figure 1d-g). We built a Generalized Linear Model (GLM) that included all 16 features. The absolute value of the coefficients for each feature indicated how much each annotation contributed to explaining the neural responses (Figure 2b). For this example electrode, auditory and language features both showed a statistically significant contribution to explaining the neural response. Among the strongest contributors were the four language features shown in Figure 2d-g. The average of all coefficients across the 339 electrodes in the temporal lobe is shown in Figure 2c, for which we note that the features with the highest averaged coefficients were the index in sentence, part of speech, and delta volume (further regions shown in Figure 9).

To better understand the contribution of the language features with the largest coefficients for the example electrodes, we plotted the neural responses for words that had different values for those features. In Figure 2d, we separated the words of a movie into those that appeared early in a sentence (quartile with lowest index in sentence, light gray) and those that appeared late in a sentence (quartile with highest index, dark gray). The average neural responses for this example electrode revealed notable differences between these two groups. Common to both groups, there was a deflection from baseline well before t=0. Words with high indices led to reduced voltages and words with low indices led to high voltages after t=0. In a similar fashion, we observed responses separated by nouns versus verbs (Figure 2e), high and low word length (Figure 2f), and high and low GPT-2 surprisal (Figure 2g). In all of these cases, words elicited neural responses across different features even as the neural responses were modulated by those features. Similar conclusions for this electrode can be drawn when considering auditory (supplementary Figure 8a) or visual features (supplementary Figure 8b).

Next, we asked whether the neural responses are due purely to language, and whether an audio and/or visual explanation can be ruled out. Across all electrodes, we found that there exist 244 ($\approx 16\%$) electrodes for which there was a significant ($p < 0.05$, Bonferroni corrected) word response, after controlling for all audio and visual features. The fraction of such electrodes per region is shown in Figure 2h and the locations of these electrodes are shown in Figure 2i.

**Sentence position modulates neural activity** The results for the example electrode in Figure 2d suggest that the position of a word within a sentence can have a strong impact on the neural responses. To systematically evaluate whether neural signals are dependent on word position, we first categorized words according to their linear position (Figure 3a), separating them into sentence *onsets*, sentence *offsets*, and sentence *midsets*, which are the words that occur in between. Figure 3a shows the neural responses from an example electrode located in left superior temporal gyrus. This electrode showed stronger responses to sentence onsets (left) compared to midsets (middle) and offsets (right). These differences were evident even in single words (raster plots, top), as well as in the average responses (bottom) and are summarized in Figure 3b, which shows the mean neural activity for onsets, midsets, and offsets in a 100ms window. Similar to our analysis in the previous section, we evaluated mean neural activity at five evenly spaced 100ms windows, starting from the word onset. The activity shown in Figure 3b was taken from the window with the most significant difference between onset, midset, and offset activity (f-test, $p < 0.05$, Bonferroni corrected). Asterisks in Figure 3b denote the significance of this difference.

It might be the case that sentence onsets could be associated with a confounding feature, such as increased volume. We therefore separately plotted the responses to words in different sentence positions for cases with high and low volume. The strong modulation by part of sentence persisted across different volume levels (Figure 3c-d). Next, in addition to volume, we considered all the 16 features that we annotated in the movie, using a GLM model as illustrated in the previous section. The feature with the third highest coefficient in the GLM model was the index in sentence (Figure 3e). Running the GLM analysis for all electrodes revealed 235 electrodes ($\approx 15\%$ of total electrodes) for

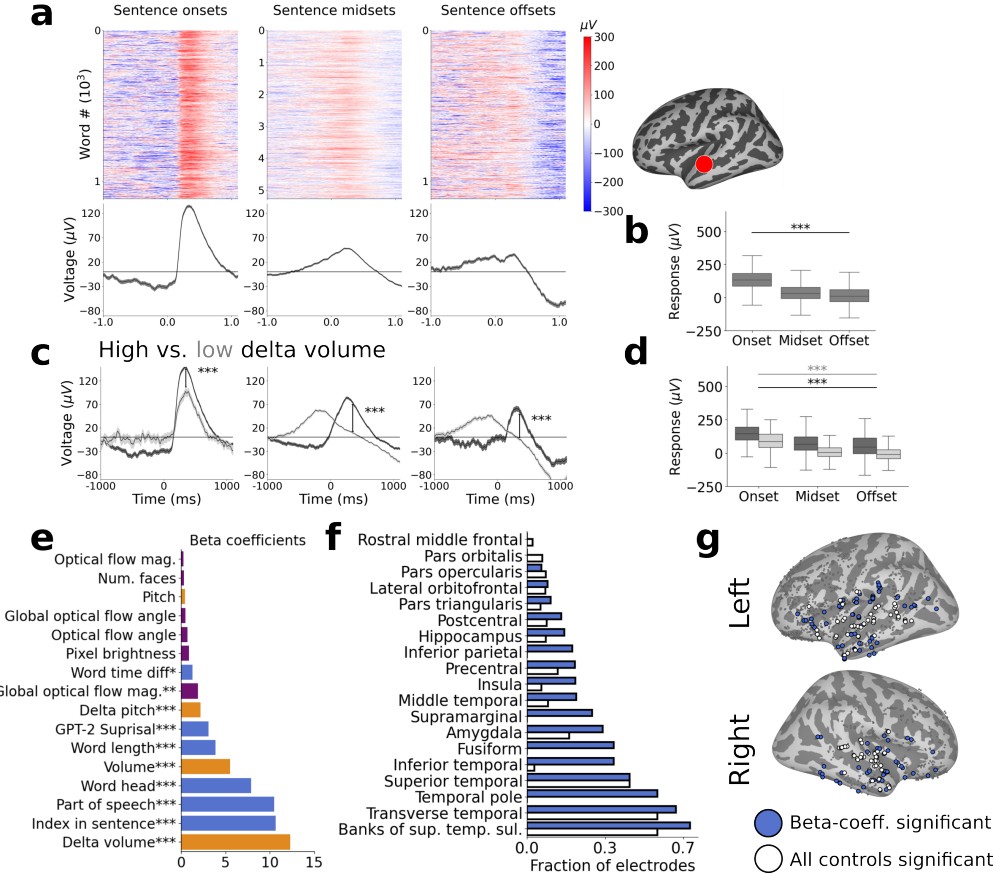

Figure 3: **Neural signals distinguish between different positions within the sentence. a.** Raster (top) and mean (bottom) neural responses for an example electrode in the left superior temporal gyrus (see electrode location on right) for words occurring at sentence onset (left), offset (right), or in between (midset, middle). The format and conventions follow Figure 2a. The box-plots (**b**) show the mean activity in a 100ms window. Asterisks show the significance of the difference between activities (f-test, Bonferroni corrected). **c.** Neural responses from the same electrode separated by trials with high volume (dark grey) or low volume (light grey). Vertical brackets and asterisks show the difference between the two conditions (two-tailed t-test). In both cases, the difference due to sentence position persists (shown by horizontal brackets and asterisks in **d**). **e.** Beta coefficients from a fitted GLM for all features, colored by category (format as in Figure 2b). Coefficients shown here are for the same electrode as in (**a**). **f.** Per region, the fraction of electrodes (shown as blue bars) in each region for which there is a significant ($p < 0.05$, Bonferroni corrected) beta coefficient for position in sentence and the fraction of electrodes (white bars) which exhibit a significant ($p < 0.05$, f-test, Bonferroni corrected) difference in activity due to sentence position after controlling for all confounds. **g.** The exact location of the electrodes from (**f**), shown as blue and white points respectively, projected onto the surface of the brain.

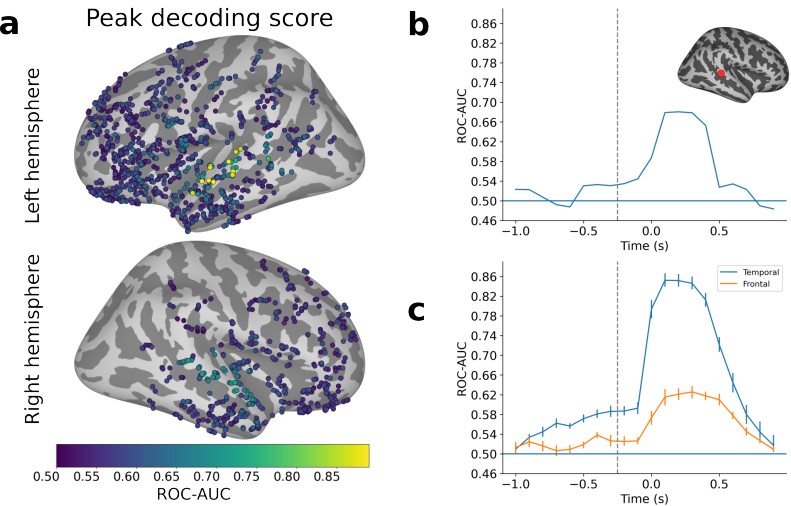

Figure 4: **Sentence onsets are linearly decodable.** A linear decoder is trained to classify portions of the movies according to whether or not a sentence onset is occurring, based on the corresponding neural activity. This decoding is done for activity in 0.25s windows, shifted in 0.1s increments, from -1s before the sentence onset to 1s after the sentence onset. The *peak* decoding performance for an electrode is the max ROC-AUC achieved across all increments. **a.** The spatial distribution of peak decoding scores. **b.** Decodability, as a function of time for an electrode in the banks of the superior temporal sulcus on the right hemisphere. **c.** The time course of decodability on the test set, for the top 10 electrodes that had the highest peak ROC-AUC score on the train set, in the temporal lobe and the frontal lobe. The test set is balanced between positive and negative examples so that chance performance is 0.5. Together, these curves reveal that for sentence onsets, information is processed before word onset enters the decoding window (dashed grey line). Error bars show a 95% confidence interval over performance per electrode. Comparing the curves reveals the mirrored time course of language processing in the frontal and temporal lobes. See supplementary Figure 6 for the same analysis, performed for word-onsets.

which the sentence position feature has a significant ($p < 0.05$, Bonferroni corrected) beta coefficient in the fitted GLMs (Figure 3f, Figure 3g blue dots).

Among these electrodes which we identified to be modulated by position in sentence, we also used a different, more stringent test to determine the influence that the position in sentence has on mean activity. For each of these electrodes, the analysis that was discussed previously with respect to Figure 3c-d was repeated for all features. Across these electrodes, controlling for all co-occurring features, revealed 114 electrodes ($\approx 7\%$ of total electrodes) that showed a significant modulation by sentence position (f-test, $p < 0.05$, Bonferroni corrected). These electrodes were predominantly located in the transverse temporal cortex and the banks of the superior temporal sulcus (Figure 3f-g).

**The temporal-course of speech decodability reveals the dynamics of language processing** We also used a linear decoder to answer questions about when and where certain language induced activity is available in the neural signal. To that end, we fitted a linear regression for every 250ms interval in a [-1000ms,1000ms] window. As discussed in the previous sections, we had observed language responses to be stronger at sentence onsets, so we first considered the case of trying to decode whether or not a sentence onset was occurring. However, the case for generic word onsets was also considered (see supplementary Figure 6), and is discussed below as well.

For each electrode, we created a training dataset of neural activity (see Appendix A.9). Positive examples consisted of sentence onsets and negative examples were taken from portions of the movie where no dialogue is occurring. We fit a regression on the train data, validate using 5-fold cross validation, and report the ROC-AUC on the test set. Figure 4a. shows the *peak* decoding performance per electrode. Here, the peak performance is the maximum performance achieved over the course of the entire considered time interval. Figure 4c shows the test-set performance per time interval in the temporal and frontal lobe, averaged across the 10 electrodes that had the highest peak decoding performance on the train set. In the frontal region, decoding peaked later than in the temporal region (300ms vs 100ms). We performed the same decoding for generic word onsets (see Figure 6). Here

we found a similar pattern as in figure Figure 4. Decoding in the temporal lobe reached a peak at 200ms, compared to 300ms in the frontal lobe.

Finally, we also attempted to linearly decode the noun vs. verb distinction in the brain (supplementary Figure 7). We saw that the noun vs. verb distinction is most decodable in the frontal lobe, where decoding performance peaks after word onset ($t = 400$ms).

# 5  Conclusion

The Brain Treebank has a unique combination of large scale, high temporal resolution, high spatial resolution, naturalistic stimuli, and many layers of manual annotation. Because naturalistic stimuli contain many uncontrolled co-occurring features, scale is critical in order to find natural experiments with controls post-hoc. We demonstrate two such an experiments: first, how response to words and sentences can be identified, even after controlling for co-occurring features, and second, how linear decoding reveals the time course of word and sentence processing. This only begins to explore what can be done with these data and annotations, and it remains to be seen what is detectable if more powerful decoding tools are applied.

**Limitations** Subjects only watched each movie once, thus one cannot simply average over repetitions of exactly the same stimulus. Although, each movie does repeat the same words and often shows the same characters, naturalistic stimuli are harder to work with than controlled experiments. Subjects all saw different movies, making the cross-subject analysis more difficult. At the same time, this means that there are more opportunities to find interesting phenomena because of the diversity of the movies that subjects saw. As with all studies that involve naturalistic stimuli, controlling for confounds can be difficult. Intracranial recordings are only possible because subjects require neurosurgery for some condition, in this case epilepsy; it is possible that this could result in some sampling bias. Additionally, the corpus includes only movies in English, although we are adding Spanish movies and subjects shortly. In this vein, we are actively working on collecting more data and hope that others who intend to collect data can collect it for the movies we have annotated here. Tools and techniques to run experiments on naturalistic data are much newer and more limited at the moment.

We have not begun to scratch the surface of the kinds of analyses possible with the Brain Treebank. For example, we have never used the speaker identities and hardly exploited multimodality, nor have we made use of the parses aside from the POS tags. We hope that the Brain Treebank will enable the development of new tools and new kinds of neuroscientific experiments at scale with natural stimuli, as well as bring the neuroscience, NLP, and linguistics communities closer together with a shared resource that has components from each.

**Acknowledgements**

We would like to thank Marcelo Armendariz for his helpful feedback and assistance with software. We would also like to thank Tergel Myanganbayar for her valuable work in preparing the code and annotations for release.

This work was supported by the Center for Brains, Minds, and Machines, NSF STC award CCF-1231216, the NSF award 2124052, the MIT CSAIL Machine Learning Applications Initiative, the MIT-IBM Watson AI Lab, the CBMM-Siemens Graduate Fellowship, the DARPA Artificial Social Intelligence for Successful Teams (ASIST) program, the DARPA Mathematics for the DIscovery of ALgorithms and Architectures (DIAL) program, the DARPA Knowledge Management at Scale and Speed (KMASS) program, the DARPA Machine Common Sense (MCS) program, the United States Air Force Research Laboratory and the Department of the Air Force Artificial Intelligence Accelerator under Cooperative Agreement Number FA8750-19-2-1000, the Air Force Office of Scientific Research (AFOSR) under award number FA9550-21-1-0399, the Office of Naval Research under award number N00014-20-1-2589 and award number N00014-20-1-2643, and this material is based upon work supported by the National Science Foundation Graduate Research Fellowship Program under Grant No. 2141064. The views and conclusions contained in this document are those of the authors and should not be interpreted as representing the official policies, either expressed or implied, of the Department of the Air Force or the U.S. Government. The U.S. Government is authorized to reproduce and distribute reprints for Government purposes notwithstanding any copyright notation herein.

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

# A   Appendix

## A.1   Audio transcription and alignment

The audio track of each movie was first annotated by commercial services (`Rev.com` and `HappyScribe.com` depending on the movie) and manually corrected by trained annotators. A custom tool was developed to refine the alignment via an auditory spectrogram of 4 seconds at a time and slowed-down audio track. Annotators were instructed to adjust the onset and offset of every word to align with the spectrogram and their perception of when the word started and ended. The audio annotation tool automatically played the audio segment corresponding to each word to allow annotators to verify their work. As the audio was played a line marked the location of the audio sample in the spectrogram in real time.

Since speech recognizers often misused or missed critical punctuation marks, these were inserted by annotators manually. Sentences were then manually segmented. Annotators were instructed not to use abbreviations, even if they are common. Annotators marked audio segments that consisted of overlapping speech or signing. These were removed from the dataset. All foreign language was marked and removed from the dataset. Annotators were instructed to transcribe literally, i.e, contractions were used in the transcript only when spoken as such. Similarly, foreshortened words, e.g., goin' vs going, were transcribed as such when used by speakers. Cardinal numbers were spelled out. Longer numbers were spelled out as spoken, including conjunctions such as "and". All overheard words were transcribed, even when they could not easily be localized on the spectrogram, for example, short words such as "to" can sometimes be heard but no specific segment of the spectrogram seems to correspond uniquely to such words. In this case annotators were asked to mark their onset and offset as they heard the words. Transcripts are as spoken, without correction, even when the speaker erred omitting a word or using a word inappropriately.

## A.2   Feature annotation

We extract 16 features that were included in the analyses (see Supplementary Table 4). Visual and auditory features are computed over a fixed 500ms window after word onset.

**Visual features**   The visual scene scalar features were extracted from the middle frame presented during a word utterance via OpenCV 4.4.0 [50]. Brightness was quantified as the average pixel HSV value channel. Flow vectors were computed as dense optical flow over grey-scale frames via the OpenCV `calcOpticalFlowFarneback` function (pyramid scale 0.5, 5 levels, window size 11, 5 iterations, pixel neighborhood of 5, and smoothing of 1.1). Number of faces per-frame was estimated via the OpenCV `CascadeClassifier` function with the Haar cascade frontal face default classifiers over gray-scale frames (scale factor: 1.1, minimum neighbours: 4).

**Auditory features**   The auditory scalar features were collected with the Python Librosa package (0.7.2) [51], an open source audio analysis library. Sound intensity and mean frequency of the audio track during word utterance were estimated, as well as their change relatively to the preceding $500ms$ window. The average intensity of the audio segment was computed as the root-mean-square (RMS) (`rms` function, frame and hop lengths 2048 and 512 respectively) of that segment. Pitch was extracted using Librosa's `piptrack` function over a Mel-spectrogram (sampling rate 48,000 Hz, FFT window length of 2048, hop length of 512, and 128 mel filters).

**Language features**   We used a state-of-the-art syntactic parser, Stanford NLP Group's Stanza [52], to parse every sentence. POS tags were recorded for every word. Surprisal was quantified as the negative-log word probability. Word probabilities were estimated by a transformer model. GPT-2 probabilities were computed via GPT-2 large using the Hugging Face Transformers 3.0.0 library [53]. Word particle surprisal were combined by summation.

All Universal Dependency features were inferred using the standard English model of the Stanza Natural Language Processing toolkit [52] and then manually corrected via a single trained annotator over the course of a year.

**Speaker annotation**   Annotators doing speaker identification were instructed to use the characters' full names, insofar as they are known. If a character is unnamed, the annotator may identify them with a brief description of their role.

Occasionally, a character had another identity that they went by. In Spider-Man: Homecoming, the AI in Peter's suit is known for more than half the movie as "suit lady," until Peter finally decides to give her the name "Karen." In such situations, the annotator marked both identities, with whichever identity they decide is primary listed first, and the secondary identity in parentheses. So, in the above example, Peter's AI is annotated as "Karen (suit lady)"

Because of our data set, we deal with quite a lot of super heroes with secret identities. If a super hero was in costume, annotators identified them by their super hero name. Out of costume, they were identified by their birth name. When they are partially in costume (say, they're in costume, but they've taken off their mask), annotators identified them by their super hero name, followed by their birth name, separated by a forward slash: e.g. Spider-Man / Peter Parker

In situations where one character is pretending to be another, the guidelines bear some resemblance to the guidelines for heroes that are partially in costume. Annotators identified them by the person being imitated, followed by the true identity of the character, separated by a percent symbol. So, for a good part of the movie Megamind, the titular character is pretending to be a museum curator named Bernard. Dialog spoken by him during these moments should be annotated as "Bernard % Megamind."

Lines that had problems and therefore that need special attention can be identified using an asterisk. Two of the most common situations where this cropped up were when multiple characters were speaking in unison, or when a "sentence" actually contains utterances from multiple characters. In the former situation, these were identified with the line with `* multiple speakers`. In the latter situation, both speakers were annotated, with an asterisk between them e.g. "Peter Parker * Tony Stark," and an asterisk was added to the line of dialog at the point where one of them stops speaking and the other begins.

### A.3   Task and stimuli

Movies were extracted from DVDs and are unchanged other than being re-encoded to a fixed frame rate (23.976 fps). Transcripts, and all annotations described in this work will be made publicly available. Due to copyrights prohibiting the release of the raw stimuli (movies) source material, multiple audio-visual sample clips and tools allowing users to verify alignment of their own movie copies will be publicly provided.

Movies were shown in full to each subject. Movies were displayed via a custom video player created in Matlab 2018b. The player ensured that the presentation was at a fixed frame rate to keep the audio and video synchronized. The presentation of movies was accompanied by regular electrical triggers sent to the neural recording system to enable accurate temporal alignment between the movie and the neural data. A 15.4 inch (resolution 2880×1800) Apple MacBook Pro Retina was placed 60-100cm in front of the subject. Subjects adjusted the volume and paused/resumed the movie as needed. The movie was paused by the experimenter any time someone entered the room or when subjects were distracted and was resumed when subjects could direct their full attention back to the movie. Subjects could freely change position, but were instructed by the experimenter, who watched the movies with the subjects, to remain focused on the stimulus or pause the movie. Subjects did not speak during the presentation of the movie nor did they overhear any other speech other than that found in the movie.

### A.4   Data acquisition and signal processing

Clinicians implanted subjects with intracranial stereo-electroencephalographic (sEEG) depth probes containing 6-16 0.8 mm diameter 2 mm long contact electrodes (Ad-Tech, Racine, WI, USA) recording Intracranial Field Potentials (IFPs) with 1.5 mm separation. Each subject had multiple such probes implanted in locations determined by clinical concerns entirely unrelated to the experiment. Data was recorded using XLTEK (Oakville, ON, Canada) and BioLogic (Knoxville, TN, USA) hardware with a sampling rate of 2048 Hz.

During movie presentation, triggers were sent to a separate channel on the neural recording device via a USB connection to a dedicated trigger box (Measurement Computing USB-1208FS) using the

Psychtoolbox 3 Matlab package. Each pulse was logged with both its wall-lock timestamp and its movie timestamp. Individual triggers were sent every 100ms. Specific events (movie start, pause, resume, and end) were marked by bursts of triggers (10, 8, 9, and 11 respectively) separated by 15ms. All triggers consisted of a 15ms electrical burst at a magnitude of 80mV. An automated tool found triggers and aligned the movie and neural data.

## A.5 Cortical surface extraction and electrode visualization

For each subject, pre-operative T1 MRI scans without contrast were processed with FreeSurfer's `recon-all` function with `-localGI`, which performed skull stripping, white matter segmentation, surface generation, and cortical parcellation [54–73]. iELVis [74] was used to co-register a post-operative fluoroscopy scan to the preoperative MRI. Electrodes were manually identified using BioImageSuite [75], and then assigned to one of 68 regions (according to the Desikan-Killiany atlas [46]) using FreeSurfer's automatic parcellation. The alignment to the atlas was manually verified for each subject. One subject (subject 5) had a large frontal lesion in the right hemisphere that prevented alignment to an atlas. Electrodes from this subject were included in all analyses except for region analyses and they were not plotted on the brain.

Corrupted signal electrodes ($n = 114$) with extensive durations of static signal recordings were manually removed from consideration prior to any downstream analysis. When determining significance, Bonferonni correction was done according to the remaining number of electrodes.

Depth electrodes in the white matter were projected to the nearest point on that boundary, and were labeled as coming from that region (for the purposes of region significance analyses). Of the 1,688 total electrodes, 1,504 of the electrodes were able to placed in this way into a particular region. The relevant region analyses are shown in Figure 2h-i, Figure 3f-h, Figure 15e-f, Figure 4b, Figure 6b, Figure 7b, Figure 10e.

This procedure is very similar to the post brain-shift correction methods used for electrocorticography electrodes [76]. For visualization purposes, all electrodes identified to lie in the gray matter or on the gray-white matter boundary were first projected to the pial surface (using nearest neighbors), and then mapped to an average brain (using Freesurfer's fsaverage atlas) for the visualizations shown in the main text.

**Example electrodes**    The electrode shown in Figure 2 is LT1bIb6 from Subject 2. The electrode shown in Figure 3 is T1b2 from Subject 3. The electrode shown in Figure 4 is T1cIf8 from Subject 10. The electrode shown in supplementary Figure 15 is T1bIc6 from Subject 1.

## A.6 Word responsiveness

To determine the word responsiveness of an electrode, we compared pre-onset windows to post-onset windows (Figure 13). Precisely, we compared the mean activity in a 100ms window before word onset to the activity in a 100ms window after word onset with a two-tailed paired t-test. The windows were separated by an interval of 1s. This test was performed for absolute offsets of $[-0.5s, -0.4s, -.3s, -.2s, -.1s]$ (Figure 13). This is done to account for the fact that any one offset may "miss" the neural response by chance (see Figure 13). An electrode is *word responsive* if at least one of the tests shows a significant (after correction for multiple comparisons) difference between pre- and post- onset activity. In such cases, we report the significance of the t-test with the lowest p-value.

## A.7 Testing difference between conditions

When determining the significance of the difference between two conditions (Figure 2c, Figure 3b, Figure 15a,c,d), we used a two-tailed t-test to compare the mean activity in a 100ms window for the two conditions. Five t-tests are performed, at absolute offsets of $[0s, 0.1s, 0.2s, 0.3s, 0.4s]$ and we say that the two conditions result in different neural responses if there exists a test for which there is a significant difference, after correction for multiple comparisons. In such cases, we report the significance of the tests with the lowest p-value. As in the above section, this is done to account for the fact that any one of the tests may miss the difference between the two conditions by chance.

## A.8 GLM Analysis

GLM analysis was done for using `pymer4` [77]. The input dataset consisted of a feature vector for each word (token) in the movies that the subject watched. The feature vector consisted of the 16 visual, auditory, and language features listed in Table 4 that co-occurred with each word. All features were normalized across the movie in which they occurred. The target was the mean neural response in a 500ms window after word onset. For subjects that watched many movies, linear mixed effect modeling was used, where the movie was taken to be a random effect.

## A.9 Linear decoding

**Model**   The model is a logistic regression.

**Data pre-processing**   Neural data is decimated by a factor of 10. Data is normalized to 0 mean and unit standard deviation. Normalization is done such that no data-leakage occurs (see below).

**Dataset**   The sentence-onset decoding task requires the model to distinguish between neural activity from an interval in the movie during which a sentence is beginning versus an interval during which no speech is occurring. To obtain positive examples, for every sentence onset, we extract 2s of neural activity, centered on the sentence onset. To obtain negative examples, we divide the movies into 3s segments, and filter for segments that do not overlap with any speech time-stamps. The size of 3s guarantees that there is at least a 500ms buffer between every positive example and every negative example (see below). The dataset is balanced so that an equal number of negative and positive examples occur. Data is drawn from all recorded movies per subject.

**Training**   We are interested in answering the question, how does decodability vary across time? To this end, we divide each example into 250ms intervals. Per each time interval, per electrode, we fit a regression. Training was done on a single NVIDIA Titan RTXs (24GB GPU Ram) with 80 CPU cores.

**Evaluation**   Per electrode, we create an 80/20 train/test split. The model performance is reported on the test set. Train/test splits are shared between electrodes in the same subject. In Figure 4b-c and supplementary Figure 5, we select the top 10 electrodes with the highest score on the train-set (5-fold cross-validation) per region, and report the performance of these electrodes on the test set. The same is done in supplementary Figure 6b,d-e and supplementary Figure 7b,d-e.

## A.10 Part of speech modulates activity

Parts of speech are of particular importance for their fundamental role in linguistics and natural language processing (NLP). Indeed, the two word classes, nouns and verbs, are widely recognized to be among the few linguistic universals [78, 79]. Part-of-speech was a significant predictor in the example electrodes shown in Figure 2 and Figure 3. Given their importance in language, we directly compared the responses to nouns versus verbs (Figure 15). Figure 15a shows the responses of an example electrode located in the left superior temporal gyrus (inset) which showed stronger responses to verbs compared to nouns.

The GLM analysis showed that there were no electrodes which exhibited activity exclusively modulated by part-of-speech. Instead, the neural activity was captured by multiple features as shown in the previous examples. Figure 15b shows that the main feature for this electrode is the index in sentence, followed by the part-of-speech and volume. Indeed, after separating the responses according to the position in the sentence, there was a small but significant difference between nouns and verbs for sentence offsets but not for sentence midsets and onsets (Figure 15c). The differences between nouns and verbs persisted across high and low volumes (Figure 15d). There were no electrodes for which a difference in part-of-speech was observed across all sub-samplings for all features. But there were 69 electheretrodes for which part of speech has a significant ($p < 0.05$, Bonferroni corrected) beta coefficient in the GLM analysis. Figure 15e shows the exact location of these electrodes and Figure 15f shows the fraction, per region, of the part of speech significant electrodes. We also found that the noun-verb distinction is linearly decodable (see Figure 7), with significant decoding performance distributed across the brain (Figure 7a), and with the highest decoding performance observed in the frontal lobe and cingulate (Figure 7b-e).

Finally, we observed a difference in the magnitude and timing of the peak neural response between nouns and verbs (Figure 17). For each electrode, we computed the mean of the neural response, averaged across all words. Restricting our attention to those electrodes which show at least a moderate neural response (Cohen's $d > 0.1$), we can compute the peak of that mean response (Figure 17b) and observe that it is lower in the case of verbs at sentence onsets ($\mu \approx 32.8, \sigma = 26.7 \ \mu V$ for verbs, $\mu \approx 36.7, \sigma = 29.4 \ \mu V$ for nouns), but higher in the case of verb midsets ($\mu = 33.4, \sigma = 23.7 \ \mu V$ for verbs, $\mu = 31.6, \sigma = 25.1 \ \mu V$ for nouns). We also find the timing (Figure 17c) of the sentence midset peaks and observe that it is later in the case of verbs ($\mu \approx 188, \sigma = 313 \ ms$ for verbs, $\mu \approx 77, \sigma = 360 \ ms$ for nouns).

# B  Supplementary figures

| Subj. | Age | Sex | Movies | Time (h) | # Sent. | # Words | # Lemmas | # Elec. | # Probes |
|---|---|---|---|---|---|---|---|---|---|
| 1 | 19 | M | 7, 18, 19 | 5.6 | 4372 | 27424 | 4489 | 154 | 13 |
| 2 | 12 | M | 2, 3, 4, 8, 9, 17, 21 | 13.5 | 9870 | 57731 | 9164 | 162 | 47 |
| 3 | 18 | F | 5, 11, 12 | 7.5 | 5281 | 31596 | 4547 | 134 | 12 |
| 4 | 12 | F | 10, 13, 15 | 3.7 | 4056 | 23876 | 4017 | 188 | 15 |
| 5 | 6 | M | 7 | 1.35 | 1282 | 7908 | 1481 | 156 | 12 |
| 6 | 9 | F | 6, 13, 20 | 2.8 | 3789 | 20089 | 3349 | 164 | 12 |
| 7 | 11 | F | 5, 13 | 3.08 | 3523 | 19068 | 2828 | 246 | 18 |
| 8 | 4 | M | 14 | 0.94 | 860 | 3994 | 537 | 162 | 13 |
| 9 | 16 | F | 1 | 1.80 | 1558 | 9235 | 1480 | 106 | 12 |
| 10 | 12 | M | 5, 16 | 3.08 | 3981 | 22147 | 3004 | 216 | 17 |

Table 2: All subjects language, electrodes and personal statistics. Columns from left to right are the subject's ID and information (age and gender), the IDs of the movies they watched (corresponding to supplementaryTable 3), the cumulative movie time (hours), number of sentences, number of words (tokens) and number of unique lemmas (canonical word forms), as well as the number of probes the subject had and their corresponding number of electrodes.

| # Movie | Year | Length | Sent. | Words | Unique words | Nouns | Unique nouns | Verbs | Unique verbs |
|---|---|---|---|---|---|---|---|---|---|
| 1 Antman | 2015 | 7027 | 1558 | 9869 | 1944 | 1358 | 705 | 1545 | 580 |
| 2 Aquaman | 2018 | 8601 | 1054 | 7233 | 1544 | 1069 | 520 | 1104 | 508 |
| 3 Avengers: Infinity War | 2018 | 8961 | 1523 | 8529 | 1750 | 1083 | 607 | 1317 | 495 |
| 4 Black Panther | 2018 | 8073 | 1254 | 7580 | 1606 | 1093 | 553 | 1209 | 508 |
| 5 Cars 2 | 2011 | 6377 | 2051 | 11407 | 2037 | 1572 | 724 | 1664 | 577 |
| 6 Coraline | 2009 | 6036 | 997 | 5433 | 1232 | 784 | 409 | 805 | 348 |
| 7 Fantastic Mr. Fox | 2009 | 5205 | 1282 | 8461 | 1864 | 1229 | 681 | 1227 | 484 |
| 8 Guardians of the Galaxy 1 | 2014 | 7251 | 1174 | 8295 | 1779 | 1096 | 603 | 1250 | 529 |
| 9 Guardians of the Galaxy 2 | 2017 | 8146 | 1290 | 9405 | 1824 | 1224 | 626 | 1370 | 532 |
| 10 Incredibles | 2003 | 6926 | 1521 | 9430 | 1954 | 1226 | 652 | 1557 | 591 |
| 11 Lord of the Rings 1 | 2001 | 13699 | 1514 | 10566 | 1998 | 1473 | 679 | 1487 | 598 |
| 12 Lord of the Rings 2 | 2002 | 14131 | 1716 | 11041 | 2065 | 1588 | 743 | 1619 | 646 |
| 13 Megamind | 2010 | 5735 | 1472 | 8891 | 1726 | 1172 | 602 | 1347 | 496 |
| 14 Sesame Street Ep. 3990 | 2016 | 3440 | 860 | 4220 | 787 | 717 | 231 | 706 | 217 |
| 15 Shrek the Third | 2007 | 5568 | 1063 | 7226 | 1590 | 977 | 568 | 1071 | 422 |
| 16 Spiderman: Far From Home | 2019 | 7764 | 1930 | 12189 | 1969 | 1459 | 668 | 1785 | 560 |
| 17 Spiderman: Homecoming | 2017 | 8008 | 2196 | 12295 | 2066 | 1583 | 777 | 1808 | 572 |
| 18 The Martian | 2015 | 9081 | 1570 | 11374 | 2192 | 1757 | 812 | 1677 | 622 |
| 19 Thor: Ragnarok | 2017 | 7831 | 1583 | 9683 | 1789 | 1195 | 599 | 1419 | 548 |
| 20 Toy Story 1 | 1995 | 4863 | 1320 | 7216 | 1510 | 1019 | 548 | 1027 | 395 |
| 21 Venom | 2018 | 6727 | 1379 | 7937 | 1513 | 897 | 507 | 1217 | 433 |

Table 3: Language statistics for all movies. Columns from left to right are the movie's ID, name, year of production, length (seconds), number of sentences, number of words (tokens), number of unique words (types), number of nouns, number of unique nouns, number of verbs and number of unique verbs.

| # | Feature | Category | Description |
|---|---------|----------|-------------|
| 1 | Pixel brightness | Visual | The mean brightness computed as the average HSV value over all pixels |
| 2 | Global optical flow magnitude | Visual | A camera motion proxy. The maximal average dense optical flow vector magnitude |
| 3 | Global optical flow angle | Visual | As above, averaged over orientation (degrees) and selected by maximal magnitude |
| 4 | Optical flow magnitude | Visual | A large displacement proxy. The maximal optical flow vector magnitude |
| 5 | Optical flow angle | Visual | The orientation (degrees) of the above flow vector |
| 6 | Number of faces | Visual | The maximal number of faces per frame |
| 7 | Volume | Auditory | Average root mean squared watts of the audio |
| 8 | Pitch | Auditory | Average pitch of the audio |
| 9 | Delta volume | Auditory | The difference in average RMS of the 500ms windows pre- and post- word onset |
| 10 | Delta pitch | Auditory | The difference in average pitch of the 500ms windows pre- and post- word onset |
| 11 | GPT-2 surprisal | Language | Negative-log transformed GPT-2 word probability (given preceding 20s of language context) |
| 12 | Word time length | Language | Word length (ms) |
| 13 | Word time difference | Language | Difference between previous word offset and current word onset (ms) |
| 14 | Index in sentence | Language | The word index in its context sentence |
| 15 | Word head | Language | The relative position (left/right) of the word's dependency tree head |
| 16 | Part of speech tag | Language | The word Universal Part-of-Speech (UPOS) tag |

Table 4: **Extracted visual, auditory, and language features used to model the neural responses**. All features were used as regressors in the GLM analysis and controls in mean activity t-tests and f-tests. The difference between 2 and 4 is that 2 is the magnitude of the averaged optical flow vector, with the average being taken over all optical flow vectors on the screen, whereas 4 is the magnitude of the largest individual optical flow vector on the screen.

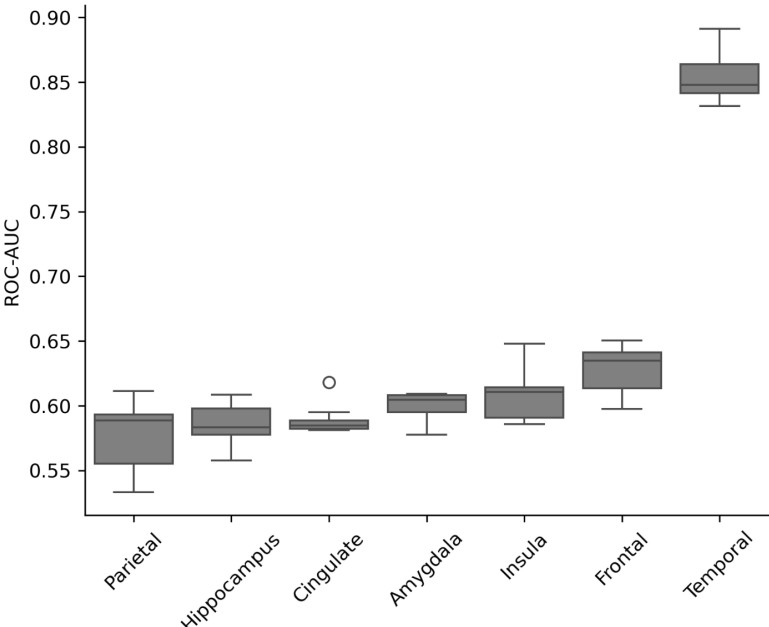

Figure 5: **Decodability of sentence onsets per region.** After decoding sentence onsets per electrodes (see Figure 4), we find distribution of the peak *test* ROC-AUC scores in each region, for the 10 electrodes in each region with the highest cross-validation ($k_{folds} = 5$) ROC-AUC on the *train* set. Boxes show quartiles and whiskers show $1.5\times$ the interquartile range. Outliers shown as points beyond the whiskers.

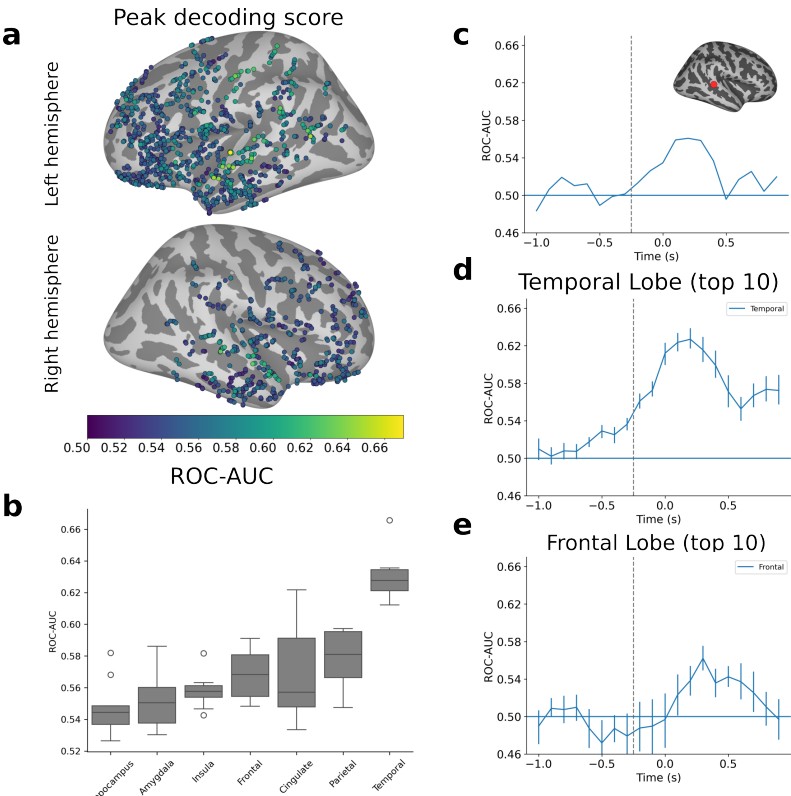

Figure 6: **Word onsets are linearly decodable and reveal the time course of language processes in the brain.** We perform the same analysis as shown in Figure 4, but for word-onsets, instead of sentence-onsets only. A linear decoder is trained to classify portions of the movies according to whether or not speech is occurring, based on the corresponding neural activity. This decoding is done for activity in a 0.25s window, which shifts in 0.1s increments from -1s before word-onset to 1s after word-onset. The spatial distribution of decoding scores, shown in (**a**) and (**b**), after a max has been taken over all windows, shows that word onsets are most decodable in the temporal and frontal lobes. Decodability, as a function of time, shown in (**c**), (**d**), and (**e**), reveal that some word onset information is processed before word onset enters the decoding window (dashed grey line). Averaging over time across the top 10 electrodes per region, as in (**d**) and (**e**), reveals the mirrored time course of language processing.

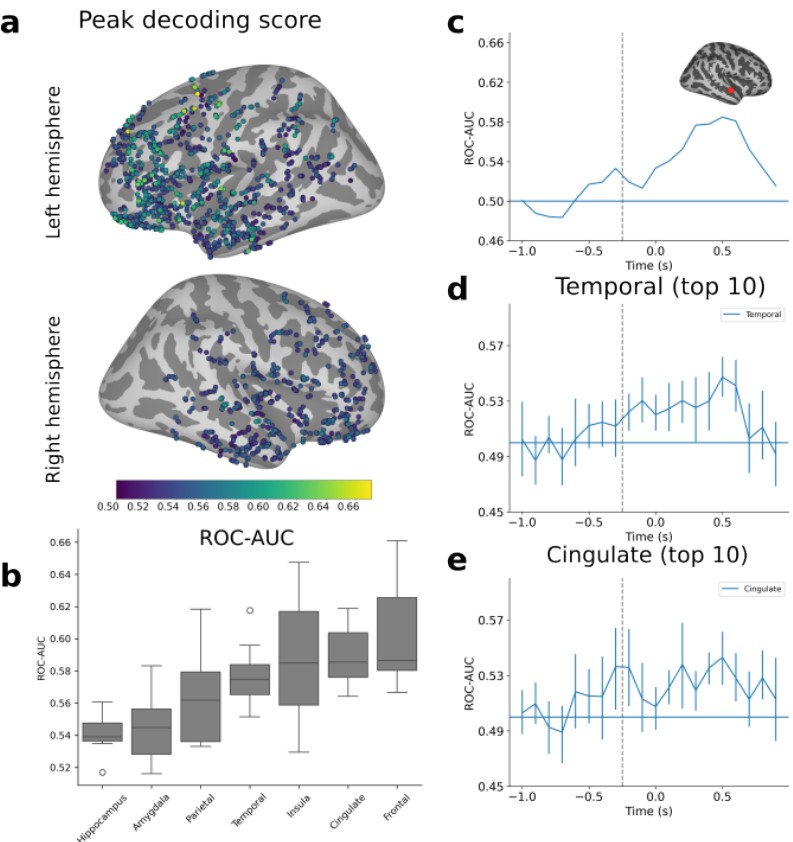

Figure 7: **Part of speech information is linearly decodable.** We perform the same analysis as shown in Figure 4 for nouns and verbs. A linear decoder is trained to classify words as either nouns or verbs, based on the corresponding neural activity. This decoding is done for activity in a 0.25s window in 0.1s increments. The spatial distribution of decoding scores, shown in (**a**) and (**b**), after a max has been taken over all windows, shows that part of speech is most decodable in the frontal, cingulate, insula, and temporal regions. Decodability, as a function of time, shown in (**c**, for an electrode in the superior temporal lobe), (**d**), and (**e**), reveal that some part of speech information is processed before word onset enters the decoding window (dashed grey line). Averaging over time across the top 10 electrodes per region, as in (**c**) and (**d**), reveals the time course of processing.

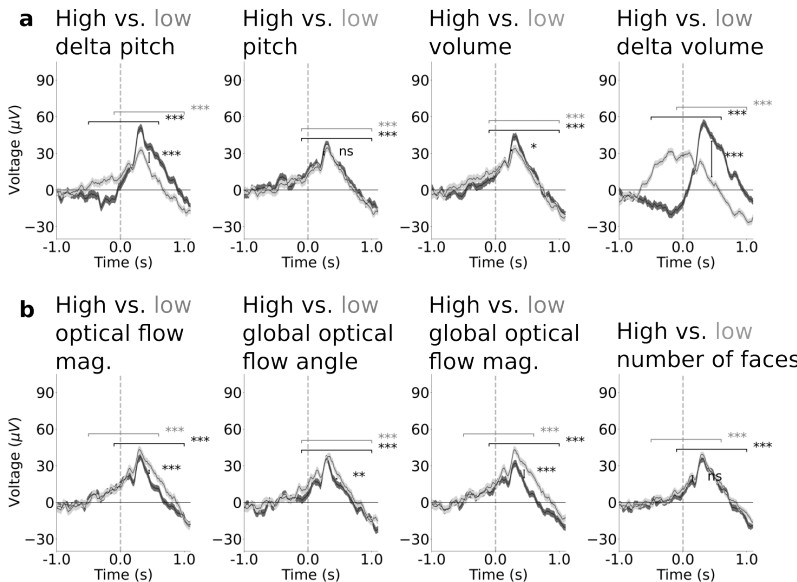

Figure 8: **Neural responses to word onsets are observable, even after controlling for visual and audio features. a**. Mean response to word onsets, after controlling for audio features for the same example electrode as shown in Figure 2. The same conventions as Figure 2c are followed. Vertical brackets and corresponding asterisks show the difference between conditions. Horizontal brackets and asterisks show the significance of the word onset response. **b.** Mean response to word onsets, after controlling for visual features. In both (a) and (b), significant response to word onset can be observed, even after controlling for audio and visual features respectively.

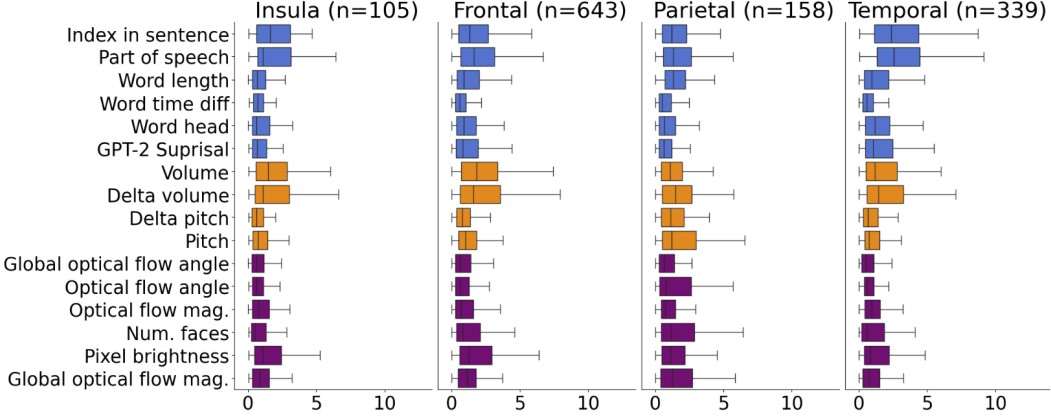

Figure 9: Magnitude of beta coefficients, averaged per region.

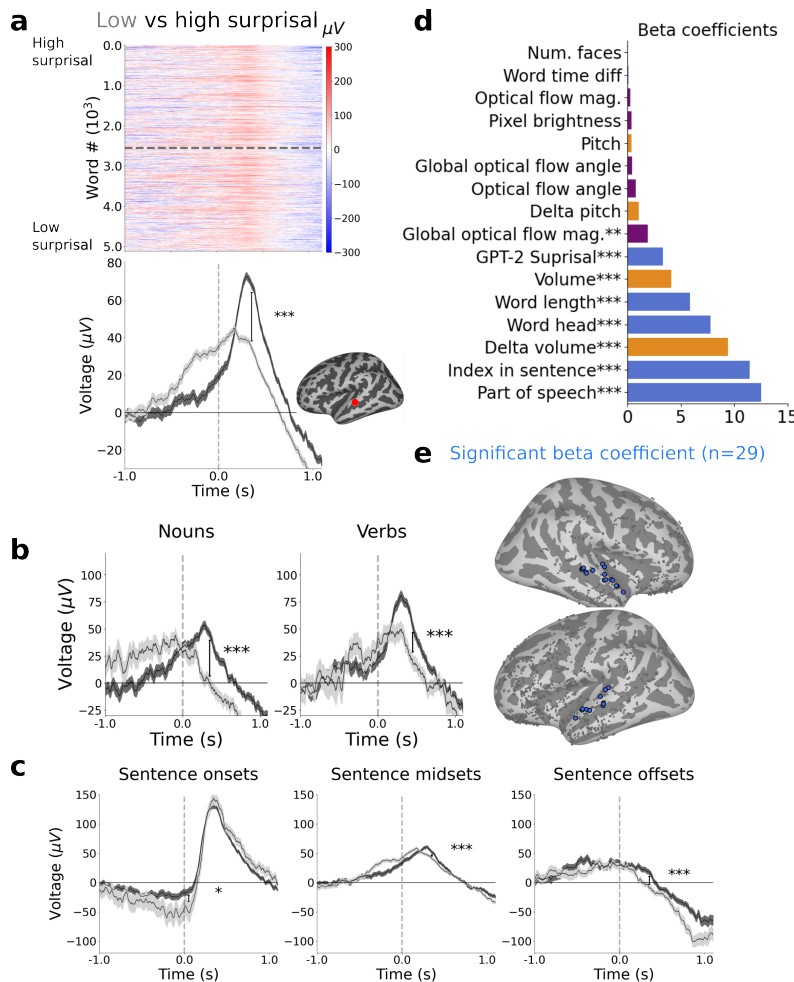

Figure 10: **Neural responses distinguish high and low surprisal. a.** Raster and mean plots aligned to word onsets for an example electrode in the right superior temporal gyrus (see inset in **d**; this is the same electrode as shown in Figure 15) separated by high and low surprisal. The difference between high and low surprisal words remains even after controlling for other features, such as part of speech (b) and position in sentence (c). GLM analysis reveals that activity in this electrode is modulated in part by surprisal, as well as by other features (d). There are 29 electrodes where surprisal has a significant beta-coefficient; these are all located in the superior temporal lobe (e).

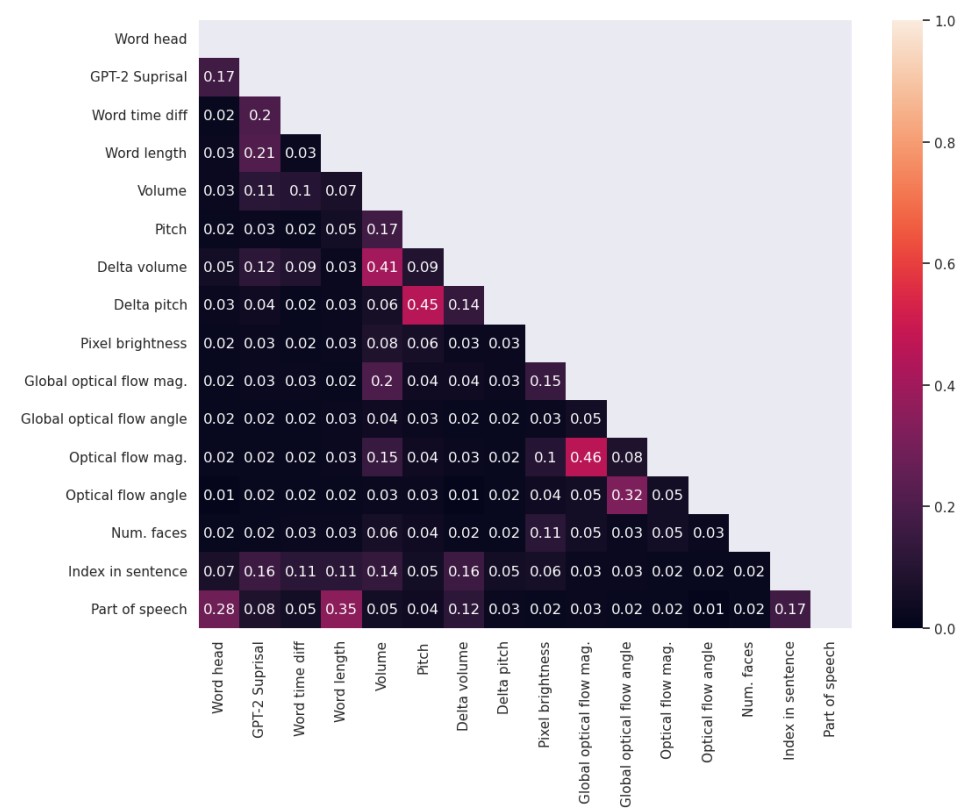

Figure 11: The (absolute value) of Pearson's *r* between input features, averaged across movies.

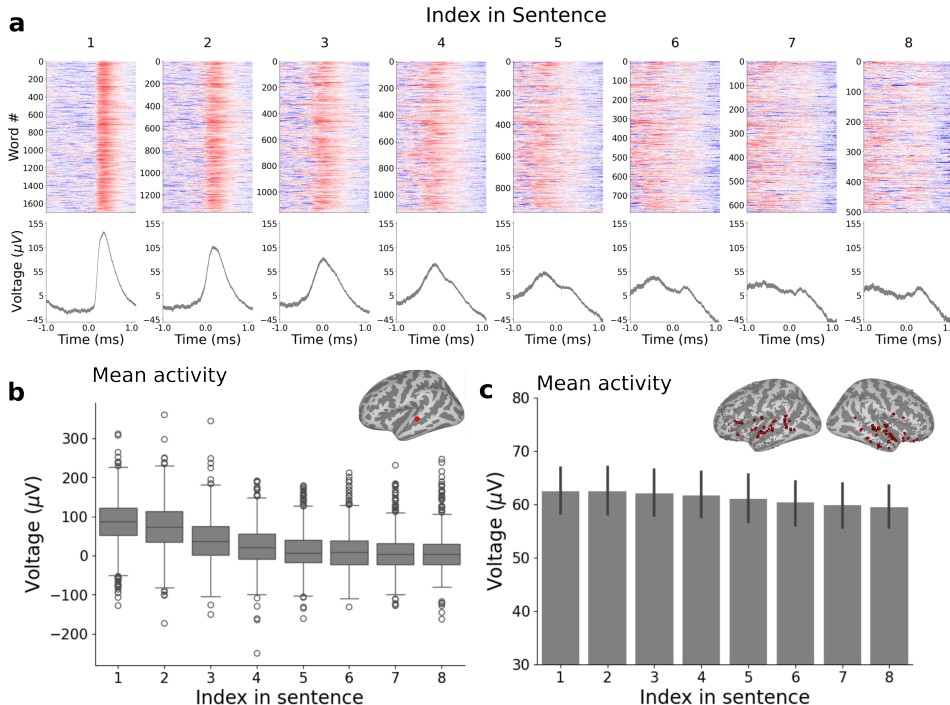

Figure 12: **Neural response decreases as a function of position in the sentence**. Making a more fine-grained examination of sentence position, we observed a trend in which mean activity decreased monotonically with the index in the sentence. (**a**) The neural response per index in sentence is shown for the first eight sentence positions for an electrode in the left temporal lobe (same electrode as shown in Figure 15). (**b**) The mean activity for this same electrode (location shown in inset) is taken for a [0ms,500ms] window after word onset. The box shows the quartiles, while the whiskers show $1.5 \times$ the interquartile range, over all words at a given position. (**c**) Taking the mean of the magnitude over this same window for all word responsive electrodes shows the same trend. Error bars show a 95% confidence interval over electrodes. A word-responsive electrode is defined, as in Figure 2, as an electrode that shows a significant difference between pre- and post- onset activity. Here we restrict our attention to those electrodes ($n = 116$, locations shown in inset) for which this difference has at least a moderate effect size (Cohen's $d > 0.1$). Note that we do not believe this result stands in opposition to previous findings, such as in [80], foremost because we consider a much different distribution of sentences in our work. The sentences shown to subjects in this work cover a wide variety of forms, and importantly, are usually part of a longer dialogue. To make a direct comparison with previous studies of sentence processing, a more fine-grained inventory of sentence types should be made over the movie transcripts.

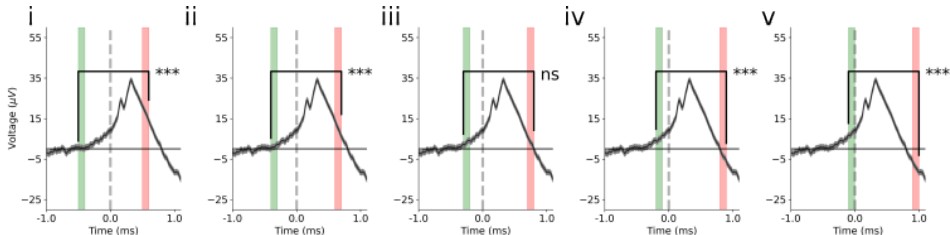

Figure 13: **Schematic of word-responsiveness testing procedure**. We test for word responsiveness at five different points (i-v). The grey line shows mean neural response, averaged across a movie. Shading shows standard error. At each point, a two-tailed paired t-test is performed between the mean activity in a pre-onset (green) and a post-onset (red) window of 100ms. We use multiple tests to account for the fact that sometimes the difference in activity may be 0 simply due to the absolute offset of the windows (this is the case for iii). We say that an electrode is word-responsive, if there is at least one test for which there is a significant difference between pre- and post- onset activity, after correcting for multiple comparisons.

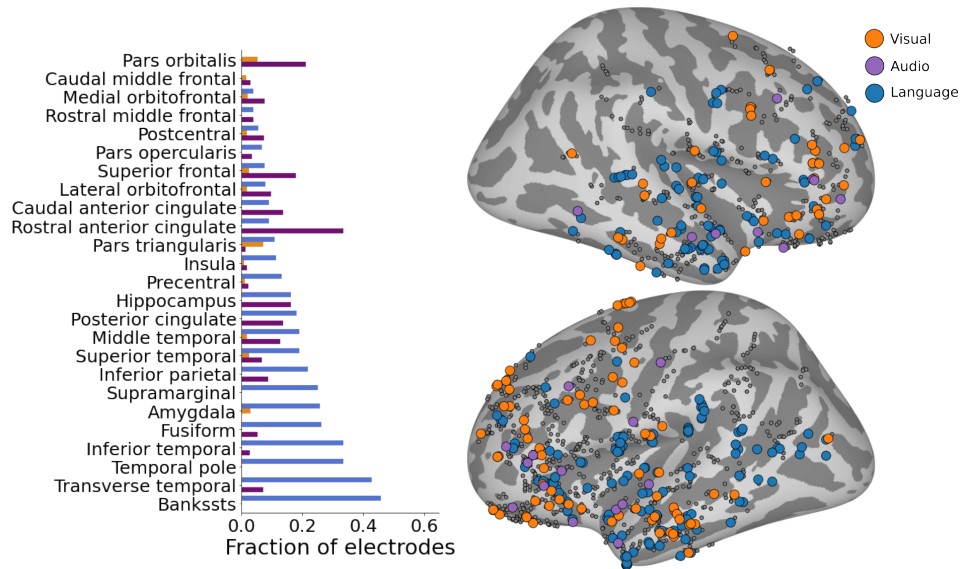

Figure 14: **Unimodal responsive electrodes**. We categorize features as either *visual*, *audio*, or *language*. For each electrode, we use the GLM analysis to determine whether a given electrode's activity has a significant (after Bonferroni correction) response for features from a single category, to the exclusion of the other categories.

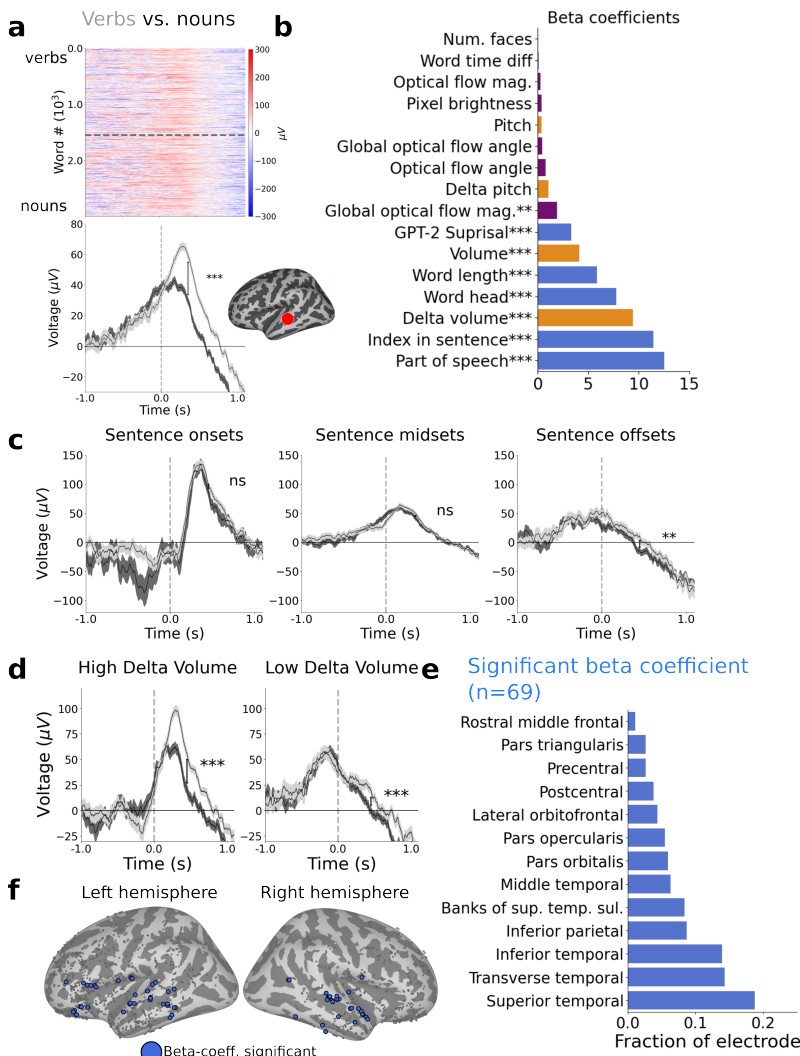

Figure 15: **Neural responses distinguish nouns and verbs. a.** Raster and mean plots aligned to word onsets for an example electrode in the left superior temporal gyrus (see inset) separated by nouns (bottom in raster plot, dark grey in mean plot) and verbs (top in raster plot, light grey in mean plot). **b.** GLM analysis reveals that activity in this electrode is modulated by part of speech, as well as by other features. **c.** For this electrode, a significant difference between nouns and verbs does not remain for the sentence onsets condition, after sub-sampling over sentence position. **d.** But, a difference does remain for all sub-sampled conditions, when controlling for other features, such as volume. Using the GLM analysis, allows us to judge the influence of part-of-speech on a per-word basis. **e.** The fraction of electrodes, per region, of electrodes where part of speech has a significant beta-coefficient (total $n = 69$); these are mainly located in the temporal and frontal lobes. **f.** The exact location of these electrodes (blue) projected onto the surface of the brain.

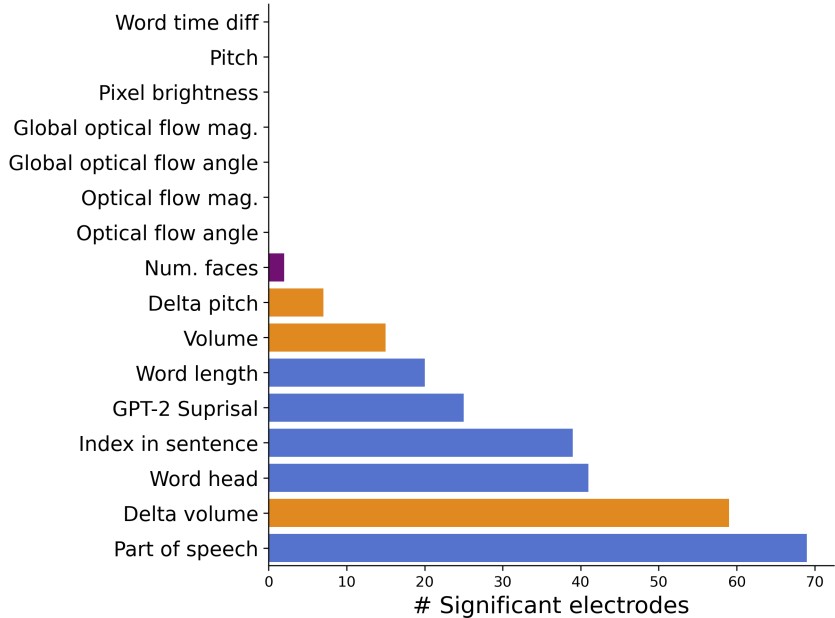

Figure 16: **The other factors which influence activity in part-of-speech-sensitive electrodes**.
An electrode is said to be sensitive to part-of-speech if a GLM fitted to mean neural activity has
a significant beta coefficient ($p < 0.05$, after corrections for multiple comparisons) for the part-of-speech feature. Among all such part-of-speech sensitive electrodes ($n = 69$), the number of electrodes
that have other significant beta coefficients is shown.

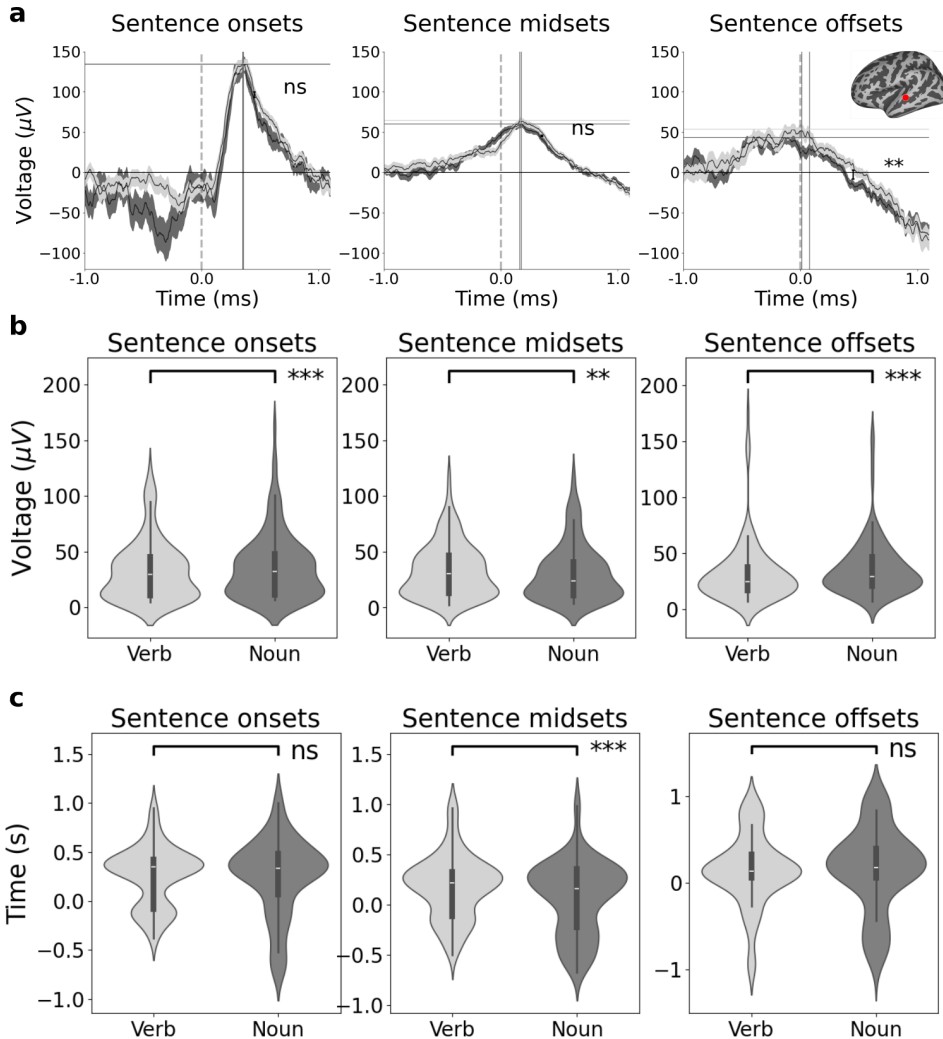

Figure 17: **Noun vs. verb peak amplitude and timing.**. For each electrode, we consider the mean signal. See, for example, (a) which shows the mean activity for an electrode in the STG (the same electrode shown in Figure 15). For an electrode, we find the amplitude (horizontal lines) of the peak mean activity and the timing of the peak (vertical lines). Across many electrodes, we observe a difference in the peak amplitudes such that nouns induce a higher response than verbs for sentence onsets, while verbs induce a higher response for offsets and midsets. The electrodes in (b) and (c) are those electrodes which respond to language (see Figure 2d), with the additional condition that the language response have at least moderate effect size (Cohen's d > 0.1).

## C  Data documentation

The brain recordings and annotations are released at the subject level, and can be thought of as the raw source, from which derivative machine learning datasets may be created. An example of a dataset derivation could be: segmenting the audio track by word boundaries and then training a decoding model to map for neural recordings to word identity. Another example could involve segmenting the recording into uniform intervals and then training a decoding model to predict average color on screen. We release the recordings in their entirety to allow for this flexibility.

The website contains the following assets:

1. `quickstart.ipynb` A quickstart IPython notebook
2. `localization.zip` Spatial position of electrodes
3. `subject_timings.zip` Wall clock time of triggers used for synchronization with movie
4. `subject_metadata.zip` Movie metadata
5. `electrode_labels.zip` Semantic ID for electrodes
6. `speaker_annotations.zip` Speaker IDs for movie audio
7. `scene_annotations.zip` Scene cut annotations for movies
8. `transcripts.zip` Pre-computed features for movies
9. `trees.zip` Universal Dependency parse trees for movie dialogue
10. `sub_<sub_id>_trial<trial_id>.h5.zip` Neural recordings in HDF5 format

## D  Responsibility, License, Hosting Plan

Authors bear all responsibility in case of privacy violations. Authors release the data under a CC BY 4.0 license.

Data will be hosted on MIT CSAIL servers and will be accessible at the url `https://braintreebank.dev/`. Backups will be kept across multiple machines. Hardware will be maintained by the MIT CSAIL Infrastructure Group: `https://tig.csail.mit.edu/`.

