# OpenReview forum: "Brain Treebank: Large-scale intracranial recordings from naturalistic language stimuli"
_NeurIPS.cc/2024/Datasets_and_Benchmarks_Track — NeurIPS 2024 Track Datasets and Benchmarks Oral_

### Official Review · Reviewer_B8dp · 2024-06-22
**Brain Treebank: Large-scale intracranial recordings from naturalistic language stimuli**

**Rating:** 8
**Confidence:** 4
**Correctness:** Yes
**Clarity:** Yes

**Review:**

Are all subjects native english speakers?
Please add the details of approval number as there are human subjects.
I would have liked some machine learning baselines in supplementary.

**Strengths:**

A challenging dataset has been presented.
A great resource for future developments in BCI.
Paper is written well.

**Additional Feedback:**

NA

**Documentation:**

Yes

**Ethics:**

Medical EEG data is collected, this would require to check ethical concerns.

**Opportunities For Improvement:**

If allowed, some machine learning experiments would be great.

**Relation To Prior Work:**

Yes

**Summary And Contributions:**

Largest dataset of intracranial recordings has been presented that features grounded naturalistic language, one of the largest English universal dependencies (UD) treebanks in general, and one of only a few UD treebanks aligned to multimodal features.
Very good analysis has been presented.
Even more details are in supplementary.

---

> ### Author Rebuttal · Authors · 2024-08-17
>
> * Subject 6 is bilingual (Speaks Spanish+English).
> * Details of the approval are discussed on line 88-89. We will add the approval number.
> * To see how our data can be used for machine learning studies, please see our parallel works, which propose (1) a transformer architecture to decode a subset of our annotations \[43\] and (2) a method for studying alignments with DNN multi-modal models \[44\].
>
> \[43\] Christopher Wang, Vighnesh Subramaniam, Adam Uri Yaari, Gabriel Kreiman, Boris Katz, Ignacio Cases, and Andrei Barbu. Brainbert: Self-supervised representation learning for intracranial recordings. In The Eleventh International Conference on Learning Representations, 2022.386
>
> \[44\] Vighnesh Subramaniam, Colin Conwell, Christopher Wang, Gabriel Kreiman, Boris Katz, Ignacio Cases, and Andrei Barbu. Revealing vision-language integration in the brain using multimodal networks. In International conference on machine learning. PMLR, 2024\.

---

### Official Review · Reviewer_ySui · 2024-07-23
**Review for 1134**

**Rating:** 6
**Confidence:** 5
**Correctness:** Yes.
**Clarity:** Yes.

**Review:**

* This paper provides relatively large-scale intracranial recordings (sEEG) that could benefit the study of naturalistic language perception and other research, such as sentiment analysis. The dataset was carefully prepared; for instance, audio and EEG alignment, audio onset annotation, and other linguistic, visual features and speech tags were also provided.

* Basic evaluations and analysis of the collected data were carried out to demonstrate solidness.

* One concern is whether this dataset could be used for visual analysis, as listed in the fourth contribution of this work. It seems that the occipital lobe is not planted. Also, sEEG itself is not dense enough for vision-related studies.

* As also pointed out by the authors, few to no common movies were shared across subjects, making analysis with a broader population challenging.

**Strengths:**

See Review.

**Additional Feedback:**

See Review.

**Documentation:**

This dataset is clearly documented.

**Ethics:**

No.

**Limitations:**

The authors discussed the potential limitations of this work.

**Opportunities For Improvement:**

See Review.

**Relation To Prior Work:**

Yes.

**Summary And Contributions:**

This paper introduces the Brain Treebank, a large-scale dataset of electrophysiological neural responses recorded from intracranial probes while 10 subjects watched Hollywood movies.

---

> ### Author Rebuttal · Authors · 2024-08-17
>
> * *occipital lobe is not planted*
>   * While the early visual area is undersampled, the late visual areas in the temporal lobe are well represented.
> * *few to no common movies were shared across subjects, making analysis with a broader population challenging*
>   * Repetitions across subjects is sparse, but within subjects, the length of data collected is large. This means that the data is most effectively used for studies that can be done at a subject level and then aggregated. For example, within a given subject, studies can be made to uncover the relative difference between classes, e.g. nouns vs. verbs. Then, analysis can be done to check whether this difference is a trend that holds across subjects.

---

### Official Review · Reviewer_8tGE · 2024-07-23
**Solid paper but no machine learning**

**Rating:** 5
**Confidence:** 4
**Correctness:** correct
**Clarity:** very clear

**Review:**

Paper is very well written and illustrated. Dataset repo is also carefully organized.

My big concern is on the relevance of this work for NeurIPS community as the paper contains actually no machine learning:
In terms of data analysis, paper comes with a GLM type of method which is standard for the neuroimaging community.
The paper does not report any results involving machine learning and does not cast the GLM method as ML (eg using cross-validation as opposed to a GLM with t-test / p-values).

**Strengths:**

- good writing
- great figures
- rich dataset

**Additional Feedback:**

add machine learning for a NeurIPS audience or consider a submission in Nature Scientific Data https://www.nature.com/sdata/

**Documentation:**

good

**Ethics:**

good

**Limitations:**

no machine learning

**Opportunities For Improvement:**

- cast a scientific question into a machine learning problem that can be addressed with these data
- show results to this question even using simple baseline ML / decoding methods to set the bar.
- i would encourage the authors to consider using the BIDS standard for intracranial data to share the data https://bids-specification.readthedocs.io/en/stable/modality-specific-files/intracranial-electroencephalography.html

**Relation To Prior Work:**

good

**Summary And Contributions:**

This work proposes a dataset of 10 subjects watching a total of 43 hours of movies while being implanted with sEEG electrodes. Data comes with electrode locations, annotations on the stimuli (word onset/offset, scene labels, POS tags, speaker id etc)

Data is publicly available from https://braintreebank.dev  and is available under the CC BY 4.0 license.

---

> ### Author Rebuttal · Authors · 2024-08-17
>
> * *In terms of data analysis, paper comes with a GLM type of method. The paper does not report any results involving machine learning and does not cast the GLM method as ML (eg using cross-validation as opposed to a GLM with t-test / p-values)...cast a scientific question into a machine learning problem that can be addressed with these data*
>   * We actually do approach analysis from a decoding/machine learning angle as well\! In particular, we train a linear decoder, sort electrodes by cross-validated performance on a train set, and report performance on a test set. We do this for sentence onsets (Figure 4), word onsets (Supplemental figure 2\) and part of speech (Supplemental figure 3).
>   * Additionally, to see how this data can be used in ML studies, please see our parallel works, which propose (1) a transformer architecture to decode a subset of our annotations \[43\] as well as (2) a method for studying alignments with DNN multi-modal models \[44\].
> * Thank you for the link to BIDS. We will either adapt this format, or ensure that all necessary information is available to reproduce it.
>
> \[43\] Christopher Wang, Vighnesh Subramaniam, Adam Uri Yaari, Gabriel Kreiman, Boris Katz, Ignacio Cases, and Andrei Barbu. Brainbert: Self-supervised representation learning for intracranial recordings. In The Eleventh International Conference on Learning Representations, 2022.386
>
> \[44\] Vighnesh Subramaniam, Colin Conwell, Christopher Wang, Gabriel Kreiman, Boris Katz, Ignacio Cases, and Andrei Barbu. Revealing vision-language integration in the brain using multimodal networks. In International conference on machine learning. PMLR, 2024\.

---

> > ### Comment · Reviewer_8tGE · 2024-08-17
> >
> > Indeed Figure 4 reports time by time linear decoding with cross-validation. Sorry of missing this.
> > Does it mean that peak latency inference across electrodes is a key ML task to solve with this dataset?
> > My core concern here is that the paper focuses on descriptive statistics about the data and I don't
> > see how this can engage the ML community towards solving more neuroscience related ML tasks.
> >
> > I see that [43] and [44] demonstrate that ML research can be done with such data but the paper is not
> > articulated about the question from these papers: predicting speech onset, classification of sound vs no sound, prediction of sEEG via some encoding model fed by stimuli representations. This would frame the paper much more like an ML benchmark which is to my understanding the purpose of this NeurIPS track. As written the paper fits very well with a journal like Nature Scientific Data https://www.nature.com/sdata/
> >
> > Based on evidence from 43 and 44 I will increase my score to 5.

---

> > > ### Author Response · Authors · 2024-08-19
> > >
> > > _Is peak latency inference across electrodes a key ML task to solve with this dataset?_
> > >   - Among other things, yes, decoding latency is an important task, and is on the critical path to other more complex tasks like decoding sentences, e.g., as in [1].
> > >
> > > _focus on descriptive statistics_
> > >   - This focus is mainly due to the fact that we are introducing the dataset formally, and want to highlight what it contains. We do present decoding results (Figure 4, supplemental figures 2 and 3),  where results could be sharpened by more powerful decoding models, as shown in our parallel works, but these efforts can only begin if the dataset is more widely known. We hope you'll support it's publication given how useful it could be to the community and the fact that this is the dataset track!
> > >
> > > [1] Tang, Jerry, et al. "Semantic reconstruction of continuous language from non-invasive brain recordings." Nature Neuroscience 26.5 (2023): 858-866.

---

### Official Review · Reviewer_PEtz · 2024-07-31
**Interesting dataset**

**Rating:** 5
**Confidence:** 4
**Clarity:** The paper is clear.

**Review:**

This is an interesting dataset, reflective of recent trends towards larger neural data.

**Strengths:**

* Care was taken to manually correct the speech alignments.
* Example analyses of word and speech onsets are convincing.

**Additional Feedback:**

I wish there was a rating between "marginally above" and "marginally below accept". I would say that because of the missing analyses to show that the ~POS and~ [redacted] dependency parses are useful make the paper just below accept. The word and speech alignment analyses are interesting, suggesting that these data could be useful at least for similar investigations.

**Correctness:**

* There are some odd references in the Related Works (e.g. for MEG, EEG, fMRI) and errors (e.g. Figure 3a claims to show a red dot in the STG, but appears to be in the posterior MTG).

**Documentation:**

Access to the dataset is sufficient and IRB is mentioned in the text.

**Ethics:**

No concerns raised.

**Limitations:**

* The POS tags and dependency parses are interesting but it would have been nice to see an analysis showing that they are meaningful in the brain recordings.
* The coverage of probes varies a lot between subjects.
* Data taken from patients may not be representative of the general population, though the data are still interesting and the patients are not explicitly being operated for speech pathologies.

**Opportunities For Improvement:**

It's a pity to include so much interesting annotation (especially the POS tags and dependency parses) but analyse these. Without any baseline analyses it raises concerns for readers that the annotations may not be useful. That said, the word and speech alignment analyses were convincing.

**Relation To Prior Work:**

Not really, it seems more like the authors had an opportunity or idea and delivered on it but did not engage deeply with related work, particularly in the dataset space -- there is a lot of work in the space right now. The framing in terms of NLP is interesting but perhaps more suited to a different audience.

**Summary And Contributions:**

Brain Treebank is a dataset of electrophysiological recordings acquired from invasive probes in 10 subjects watching movies while waiting for seizures. On average, the dataset includes 4.3 hours of data per subject (a total of 21 movies were watched). Audio tracks were manually annotated for word and sentence onsets. Part of speech tagging and dependency parses were also included for the movie transcripts.

---

> ### Author Rebuttal · Authors · 2024-08-17
>
> * *Part of speech analysis and dependency analysis*:
>   * Part of speech analysis can be found in the appendix. In short, we find nouns/verbs to be linearly decodable (Supplemental figure 3\) and we also find 83 electrodes where the noun/verb feature has a significant (Bonferroni corrected) beta-coefficient in the GLM analysis (Supplemental figure 12). As for analysis for dependency parses, these would make a good direction for future work\!
> * *The coverage of probes varies a lot between subjects. Data taken from patients may not be representative of the general population*
>   * It is a trade-off. Electrophysiological recordings from invasive electrodes have the best temporal resolution, but can only be collected in clinical settings. The issues mentioned are common with intracranial electrode data, since implanting the electrodes is an invasive procedure.
> * *Figure 3a inset*
>   * The red dot could look oddly placed due to the fact that it is plotted on an inflated brain. We will mention this in the text.

---

> > ### Comment · Reviewer_PEtz · 2024-08-19
> > **Anatomical correctness**
> >
> > I appreciate that the brain surface in Figure 3 is inflated. Does that mean you're doubling down on the red dot being correct?
> >
> > If you insist on this being STG then please provide more evidence. For example, show it on the non-inflated brain with the atlas label overlaid. But I think that will fail, unless somehow the rendering is causing extreme distortions, with implications for interpretability.
> >
> > On this viewing, I would say the red dot covers the posterior inferior temporal gyrus / middle temporal gyrus.

---

> > ### Comment · Reviewer_PEtz · 2024-08-19
> > **No treebank analysis in "treebank" titled paper**
> >
> > > As for analysis for dependency parses, these would make a good direction for future work!
> >
> > It does seem like a significant omission not to include any analyses that make use of the "treebank" in the paper's title.

---

> > > ### Author Response · Authors · 2024-08-19
> > > **Treebank analysis**
> > >
> > > We agree with your sentiment! And for this initial effort, we did make use of part of the treebank. A treebank like the Penn Treebank consists of POS tags per word, relationships between words, and tags for those relationships. We made use of the POS tags in the analysis in the paper. Beyond this, understanding whole parses in the brain is a long standing problem in the field which requires significant additional work, not a quick aside in an otherwise unrelated publication. And this work can't be done without this publication because there just aren't other treebanks at any reasonable scale that are paired with neutral recordings. The dataset track is all about enabling new work with bold new datasets. We also want to see this with happen. And it can only happen with datasets like this one. We hope you'll support it's publication given how useful it could be to the community and the fact that this is the dataset track!

---

> > ### Author Rebuttal · Authors · 2024-08-19
> >
> > _Inset figure_
> > - Hi! You are indeed correct. We made a mistake. Thank you for pointing this out. In short, the inset shows the plot for the wrong electrode. When you first pointed the oddness out, we quickly checked the file with all the region labels, but failed to go further. Your second comment prompted us to actually re-create the plot again. Thank you for insisting. You can see the corrected inset in the attached pdf. We will make another careful pass for typos.

---

### Author Rebuttal · Authors · 2024-08-29

We thank the reviewers for their time spent reading our paper and giving feedback. Here, we summarize our responses to a few common reviewer points.

We were glad that reviewers found our data to be “carefully organized” (reviewer 8tGE) and “a great resource for future developments in BCI” (reviewer B8dp). We were surprised that a few comments mentioned that our work may even be too data-focused! Some comments called for decoding baselines and more evidence that this data is of general interest to the machine learning community. We are sympathetic to these comments, and to this end, we would like to point out:
1. That we do include linear decoding baseline results (Figure 4, Supplemental Figures 2 and 3).
2. In parallel work, we have shown how this data can be used for machine learning studies. Namely, we have shown its use in Transformer-based representation learning [43] and studying the alignment between machine and neural processing of multi-modal data [44]. In this work, we focus specifically on the description of our full annotation set. We agree with reviewers that there is a lot of machine learning work that remains to be done, but all of it cannot be contained in this initial presentation of the data! We believe that the dataset track is the right place to disseminate this data so that future machine learning works can continue to build on it.

## References
[43] Christopher Wang, Vighnesh Subramaniam, Adam Uri Yaari, Gabriel Kreiman, Boris Katz, Ignacio Cases, and Andrei Barbu. Brainbert: Self-supervised representation learning for intracranial recordings. In The Eleventh International Conference on Learning Representations, 2022.386

[44] Vighnesh Subramaniam, Colin Conwell, Christopher Wang, Gabriel Kreiman, Boris Katz, Ignacio Cases, and Andrei Barbu. Revealing vision-language integration in the brain using multimodal networks. In International conference on machine learning. PMLR, 2024.

---

### Decision · Program_Chairs · 2024-09-26

**Decision:**

Accept (Oral)

**Comment:**

Summary:

This paper introduces the Brain Treebank, a dataset of electrophysiological responses recorded using intracranial electrodes while subjects watched Hollywood movies. The audio was transcribed and then automatically annotated for POS and dependencies using the Universal Dependencies formalism.

Contributions:

a. a new dataset with several hours of human electrophysiological responses recorded using sEEG and then annotated for syntactic information using UD.

Summary of reviewers' opinions:

The reviewers were generally impressed with the dataset but raised a number of qustions.

Reviewer PEtz wondered whether the POS and UD annotations have been checked to nsure they are meaningful. He/she also pointed out an error in Figure 3, which was corrected during the author response phase. And they questioned the extent to which intercranial recordings generalize across subjects.

Reviewer  ySui pointed out that little movie material was shared between subjects, making response comparison difficult.

Summary of rebuttal:

The authors provided useful clarifications and quickly corrected issues raised by the reviewers.

Summary of strengths:

a. An exciting new resource, which could greatly expand the use of brain data in AI.

Summary of weaknesses:

a. Intercranial recordings may be very subjective.

b. More details about the annotation process could have been provided.

Summary opinion:

This is an important new resource, created in a careful way and that could be very beneficial to the AI / Cogsci community.